# Atypical functional connectome hierarchy in autism

Seok-Jun Hong [1,2], Reinder Vos de Wael[1], Richard A.I. Bethlehem [3], Sara Lariviere[1], Casey Paquola[1], Sofie L. Valk[4,5], Michael P. Milham[2,6], Adriana Di Martino[7], Daniel S. Margulies[8], Jonathan Smallwood[9] & Boris C. Bernhardt[1]

One paradox of autism is the co-occurrence of deficits in sensory and higher-order socio-cognitive processing. Here, we examined whether these phenotypical patterns may relate to an overarching system-level imbalance—specifically a disruption in macroscale hierarchy affecting integration and segregation of unimodal and transmodal networks. Combining connectome gradient and stepwise connectivity analysis based on task-free functional magnetic resonance imaging (fMRI), we demonstrated atypical connectivity transitions between sensory and higher-order default mode regions in a large cohort of individuals with autism relative to typically-developing controls. Further analyses indicated that reduced differentiation related to perturbed stepwise connectivity from sensory towards transmodal areas, as well as atypical long-range rich-club connectivity. Supervised pattern learning revealed that hierarchical features predicted deficits in social cognition and low-level behavioral symptoms, but not communication-related symptoms. Our findings provide new evidence for imbalances in network hierarchy in autism, which offers a parsimonious reference frame to consolidate its diverse features.

[1] Multimodal Imaging and Connectome Analysis Laboratory, McConnell Brain Imaging Centre, Montreal Neurological Institute and Hospital, McGill University, H3A2B4 Montreal, Canada. [2] Center for the Developing Brain, Child Mind Institute, 10022 New York, NY, USA. [3] Autism Research Centre, Department of Psychiatry, University of Cambridge, CB28AH Cambridge, UK. [4] Institute of Systems Neuroscience, Medical Faculty, Heinrich Heine University, 40225 Düsseldorf, Germany. [5] Institute of Neuroscience and Medicine (INM-7: Brain and Behaviour), Research Centre Jülich, 52425 Jülich, Germany. [6] Center for Biomedical Imaging and Neuromodulation, Nathan Kline Institute, 10962 Orangeburg, NY, USA. [7] Autism Center, Child Mind Institute, 10022 New York, NY, USA. [8] Frontlab, Institut du Cerveau et de la Moelle épinière, UPMC UMRS 1127, Inserm U 1127, CNRS UMR 7225 Paris, France. [9] Department of Psychology, University of York, YO10 4PH Heslington, UK. Correspondence and requests for materials should be addressed to S.-J.H. (email: sjhong@bic.mni.mcgill.ca) or to B.C.B. (email: boris.bernhardt@mcgill.ca)

Autism spectrum disorder (ASD) is a persistent neurodevelopmental condition[1] with a highly complex behavioral phenotype. Its conceptualization has undergone several transformations during the past decades. Early approaches highlighted atypical sensory processing[2,3], while more recent accounts focused on deficits in high-level cognitive and social functions, including impairments in Theory of Mind[4] and predictive abilities more generally[5]. Despite ample research on the brain basis of autism, a neurobiological framework to consolidate the co-occurrence and interplay of low- and high-level functional abnormalities remains to be established[6,7].

Classical neuroanatomy and brain imaging have provided convergent support for the emergence of brain network hierarchy throughout neurodevelopment, an architecture thought to guide the propagation of sensory inputs along multiple cortical relays into transmodal regions. This architecture is assumed to support the integration of abstract concepts, cognition, and behavior[8]. Network analyses of brain structure and function have confirmed overarching hierarchical principles of connectome organization. Specifically, studies have suggested a distinction between a network periphery containing sensory and motor regions with more locally clustered connectivity on the one hand, and a rich-club core that aggregates long-range connections and serves as a backbone for transmodal integration on the other hand[9]. This overarching system is thought to facilitate abstract, higher-order cognitive functions by helping segregate information that reflects the processing of the immediate environment from more self-generated operations emerging in transmodal, integrative cortices[10]. As ASD is linked to deficits both in sensory processing and high-level functions such as Theory of Mind, the current work evaluated whether alterations in the macroscale hierarchy could provide a parsimonious account of the diverse symptoms associated with this condition.

Resting-state functional magnetic resonance imaging (rs-fMRI) analysis offers a non-invasive means to describe macroscopic functional networks in a reproducible manner. Numerous rs-fMRI studies have focused on the mapping of connectional anomalies in ASD, but so far there is little agreement in the location of findings and nature of connectivity changes[11–14]. Such inconclusive patterns may be attributable to a number of factors, including differences across studies with respect to the specific autism populations studied, the functional systems examined, the analytic approaches employed. Effects of head motion during image acquisition and signal quality[15], as well as inconsistent image processing procedures[16], have also been noted as potential sources of variation. Despite these challenges, rs-fMRI offers a unique lens to study the interplay across different functional systems and to assess network-level principles that discriminate peripheral and core systems within the brain. Particularly, recent studies based on non-linear connectome compression techniques have revealed a principal gradient of connectivity differentiation along the cortical surface in a large group of healthy individuals[17]. In contrast to clustering-based decompositions of the brain into discrete communities[18] or recently developed connectivity boundary mapping techniques[19,20], cortex-wide gradient mapping techniques describe a continuous coordinate system at the systems level that places sensory and motor networks on one end and default mode networks (DMN) on the other end, in line with a spatial hierarchy established in earlier primate tract-tracing work[8]. Notably, a complementary simulation approach developed earlier by other groups, so-called stepwise functional connectivity analysis (SFC)[21], has shown that this gradient can also be understood as a sequence of steps in connectivity space. Indeed, SFC analysis in healthy individuals has revealed a consistent connectivity evolution from primary sensory to DMN systems that recapitulate functional connectivity gradients derived from unsupervised connectome compression[22].

Our study investigated the hypothesis that both low-level and high-level symptoms of ASD might emerge from disturbances in macroscale cortical hierarchy. We used a novel combination of connectome gradient and SFC analyses, which offer a complementary characterization of hierarchical anomalies in ASD. In fact, gradient mapping results from an unsupervised dimension reduction that visualizes spatial trends in connectivity variations following the putative cortical hierarchy, while SFC is initiated from a-priori defined sensory seeds to map connectivity transitions from peripheral to core nodes. In particular, we explored whether transmodal association cortices functionally shifted towards sensory areas in ASD, a pattern that would make sensory input harder to ignore, and that would compromise higher-order cognition by preventing the segregation of internally-driven cognitive processes.

Our analysis of a large multi-centric dataset of individuals with ASD and typically-developing controls[23,24] revealed perturbed functional gradients in autism, showing reduced functional distance between transmodal and unimodal regions. Notably, while ASD displayed an initial acceleration in SFC from sensory to early heteromodal systems, transitions into transmodal regions were aberrant, and failed to converge on the DMN core. Gradient and SFC findings in ASD were also contextualized with graph theoretical metrics, specifically connectivity distance and rich-club features[9], offering a complementary viewpoint on cortical hierarchy i.e., the dissociation of a transmodal core that aggregates long-distance connections from peripheral networks with mainly short-range connectivity. Atypical gradients and stepwise connectivity in ASD related mainly to a selective disruption in long-range connectivity, co-occurring with a deficit to fully activate the rich-club. Finally, supervised pattern learning successfully leveraged hierarchy features to predict symptom severity in social cognition and repetitive behavioral symptom domains, as indexed by the ADOS (Autism Diagnostic Observation Schedule[25]) in individuals with ASD. Effects were reproducible across included sites, in an independent validation dataset, and with respect to several MRI-specific processing choices. Taken together, our study shows that ASD is characterized by perturbations in macroscale cortical hierarchy which provides a parsimonious account of its paradoxical combination of low-level and higher-order symptoms.

## Results

**Data samples.** We studied two independent subsamples from the openly-shared Autism Brain Imaging Data Exchange initiative (ABIDE-I and II; http:/fcon_1000.projects.nitrc.org/indi/abide)[23,24]. Our discovery dataset originally included 143 individuals with ASD and 144 age-matched healthy controls, aggregated across those sites from ABIDE-I that included both children and adults (i.e., PITT, NYU, USM). MRI quality and preprocessing were visually and quantitatively evaluated. Excluding cases with low-quality structural MRI or inaccurate cortical surface extraction (visually checked and manually corrected by SLV and BCB) or high head motion on rs-fMRI (mean framewise displacement >0.3 mm, 2 SD from the mean across all subjects) resulted in a final sample of 103 ASD and 108 controls (mean ± SD age in years for ASD/controls = 20.8 ± 8.1/ 19.2 ± 7.1). Quality indices for structural and functional MRI data did not differ between ASD and controls (two-tailed Student's t-test: $p > 0.27$, $t = 1.08$ for cortical surface extraction, $p > 0.28$, $t = 1.07$ for head motion). Details on subject inclusion and quality control are provided in the Methods and Supplementary Figure 1. Details on the independent replication dataset can be found in the Methods.

**Altered macroscale gradients in autism**. We applied diffusion map embedding[26], an unsupervised non-linear dimensionality reduction algorithm, to cortex-wide functional connectomes derived from resting-state fMRI in each individual. The first principal gradient explained 24% of connectome variance in our dataset (with similar variance explained in ASD and controls across gradients, two-tailed Student's $t$-test: $t = 0.87$ $p > 0.38$; Supplementary Figure 2), and showed a gradual axis of connectivity variations that placed low-level sensory systems on the one end and the transmodal DMN on the other end, with intermediary networks in between (Fig. 1a), replicating recent data in healthy adults[17].

Following alignment of subject-wise gradients to a group-wise template from all participants[27], we compared gradient scores between ASD and controls using surface-based linear models that controlled for effects of site and age. Globally, the cortex-wide gradient was suppressed in ASD showing that both extremes (i.e., sensory networks and DMN) were contracted relative to the control range, while those in the middle axis increased (Fig. 1b). Vertex-wise comparisons (after multiple comparison correction at a family-wise error of $p_{FWE} < 0.05$) revealed decreases in ASD in transmodal medial prefrontal cortex (mPFC) and posterior cingulate/precuneus (PCC/PCU), together with increases in early integration regions at the interface between sensory and transmodal regions, such as the occipito-temporal (OT) and middle posterior temporal (pMTG) areas (Fig. 1c). Similar

findings, albeit with reduced significance, were achieved with non-parametric permutation tests[28] (Supplementary Figure 3, see Methods for details). We applied a well-established functional community decomposition to summarize surface-wide findings[18], and found reductions in the DMN core network (false discovery rate $q_{FDR} < 0.05$), whereas sensory-motor communities showed marginal increases ($p_{uncorrected} < 0.05$; Fig. 1d).

Notably, seed-based functional connectivity analysis centered on clusters of significant gradient alterations revealed predominantly connectivity reductions, not increases (Supplementary Figure 4). Indeed, mPFC and PCC/PCU showed decreased connectivity to other DMN regions, while unimodal convergence regions such as OT and pMTG displayed reduced connectivity to primary sensory and somatomotor cortices.

We did not observe any difference between ASD and controls for the second gradient (which explained an additional 13% variance), indicating the deficit is specific to the principal gradient. Furthermore, differences in the first gradient remained significant when controlling for the second gradient at each vertex (Supplementary Figure 5). While effects in gradient score changes varied across sites, findings were of consistent direction in each site (Cohen's d for PITT/NYU/USM for the decreased gradient in mPFC/PCC/PCU = $-0.55/-0.24/-0.97$; increased gradient in OT/pMTG: d = 0.37/0.30/0.78). Furthermore, although adults showed more pronounced gradient reductions in the DMN (Cohen's d children/adults = 0.29/0.88), findings

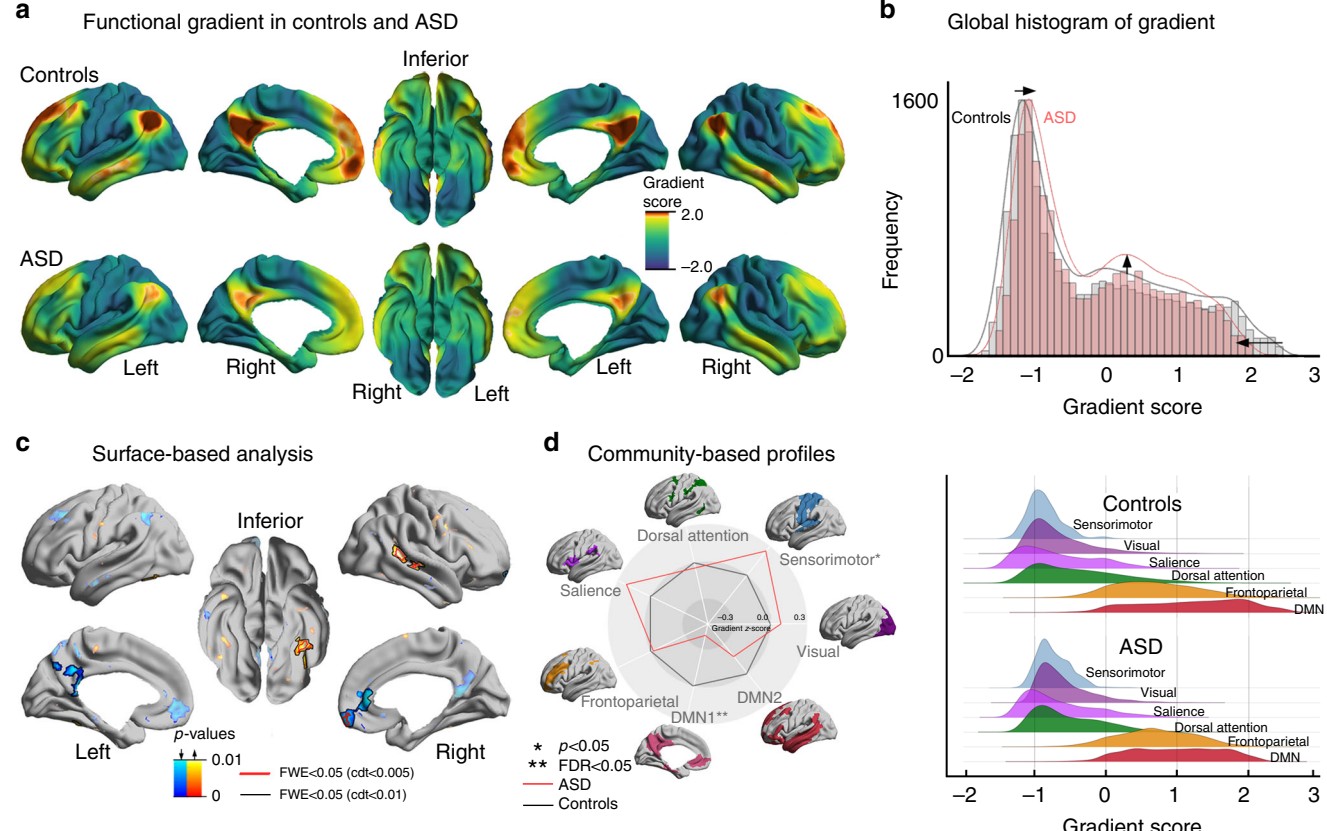

**Fig. 1** Connectome gradient mapping in autism spectrum disorders (ASD) and neurotypical controls. **a** The principal gradient in controls describes a continuous coordinate system that runs from unimodal (*dark turquoise*) regions one end to transmodal regions on the other end (*sienna*), peaking in the default mode network (DMN). Regions with similar connectivity patterns show similar coloring. In ASD, while the gradient is overall similar, one readily appreciates lower values in the DMN core. **b** Global histogram analysis confirmed that extreme values were suppressed in ASD compared to controls, while those in the mid-range increased. **c** Surface-wide statistical comparisons between controls and ASD, with increases/decreases in ASD shown in red/blue. Findings were obtained using surface-based linear models implemented in SurfStat. **d** Community-based z-score analysis of gradient score (with respect to controls) shows significant reductions primarily in DMN both spider (left) and joy (right) plots

were similar in children and adults, particularly in early integrative areas (Cohen's d children/adults in OT and pMTG = 0.73/0.92 Supplementary Figure 6). Finally, gradient findings were confirmed in an independent sample from the second ABIDE data release (ABIDE-II) ($p_{FWE} < 0.05$, Supplementary Figure 7). Our results were also robust when controlling for a number of confounding features of rs-fMRI processing, including connectivity matrix thresholding[29], global signal regression (GSR)[30], and whole-brain connectivity shift[31] (see Methods; Supplementary Figures 8-10). Specifically, altered gradient patterns in ASD were virtually identical when using GSR-preprocessed data and when analyzing differently thresholded connectivity matrices (5–25%). Results were also similar when controlling for average whole-brain connectivity strength, suggesting no strong influence of overall connectivity shift. Together, these control analyses confirmed that macroscale integration and segregation is deficient in autism relative to controls.

**Relation to stepwise functional connectomics**. Having demonstrated that the macroscale connectome gradients are perturbed in ASD, we then examined whether this relates to atypical transitions along the cortical functional hierarchy. Stepwise functional connectivity (SFC) analysis iteratively tracked connectivity from primary sensory areas (Fig. 2). Consistent with prior studies, we seeded from V1, A1, and S1[21]. In controls, we observed SFC convergence in the DMN after passing through intermediary networks, replicating earlier findings in healthy adults[21], after approximately 100–120 steps. Compared to controls, however, ASD presented with distorted SFC that did not converge in the DMN, even after 200 steps. Interestingly, delays at later SFC steps in ASD co-occurred with a paradoxical acceleration of transitions at early steps, characterized by more rapid activations of sensorimotor communities, salience, and attention networks ($q_{FDR} < 0.05$). Findings were similar when analyzing GSR-processed data, when using different thresholds for the connectivity matrices, and when controlling for average whole-brain connectivity strength

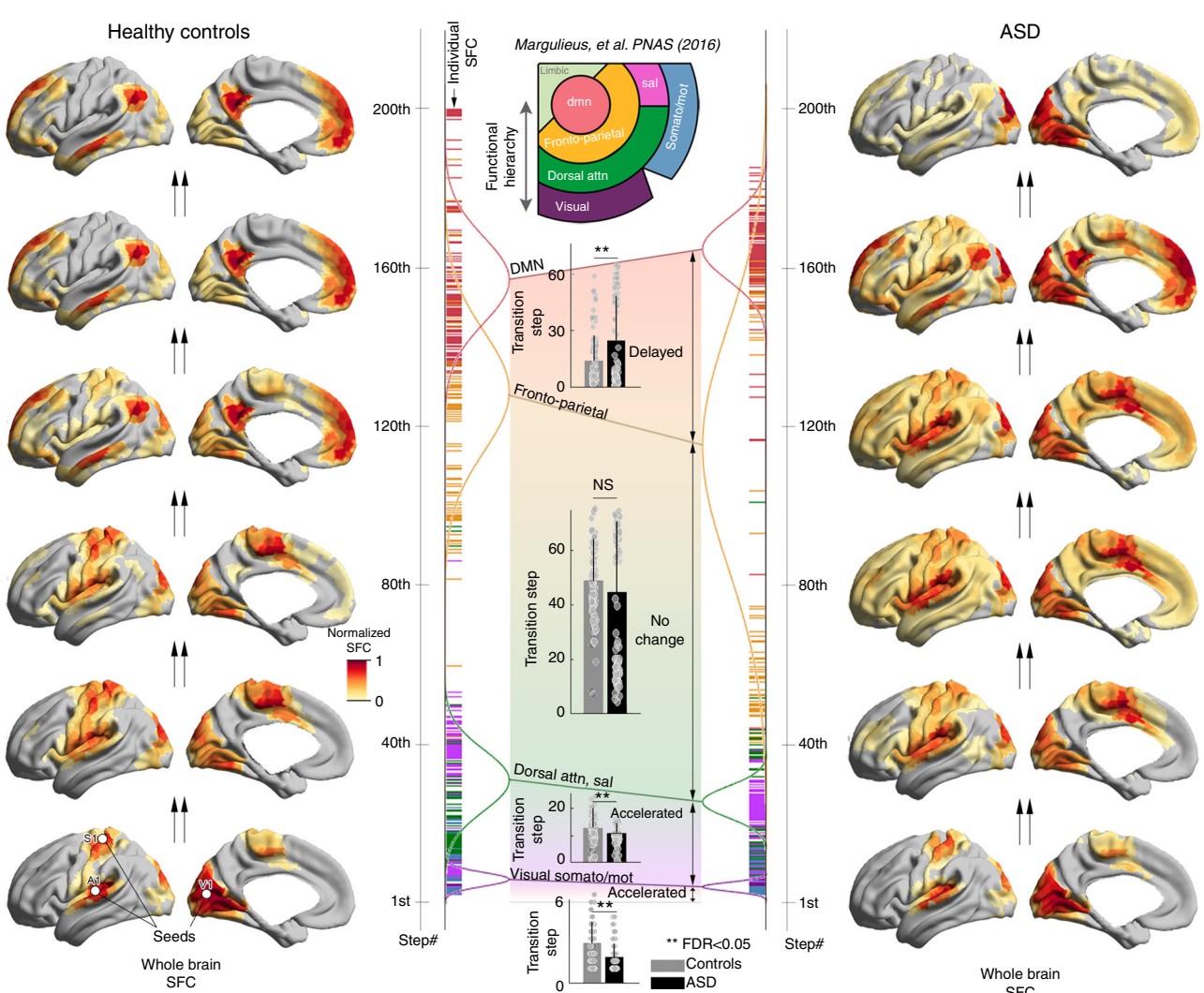

Stepwise functional connectivity difference between controls and ASD

**Fig. 2** Stepwise functional connectivity (SFC) analysis. SFC analysis combining three sensory seeds simultaneously (V1, A1, S1) mapped the human multimodal integration network in controls (left) and ASD (right). The number of connectivity steps increases moving up in the plot. In controls, transmodal DMN regions became selectively activated, after around 100–120 steps. In ASD, despite rapid initial activation of a rather extended territory, a selective DMN core activation did not occur in a comparable step range as in controls. The middle panel shows SFC findings stratified across four hierarchy levels, adapted from Mesulam[8] and Margulies et al.[17]. The bar represents a mean ± 1 SD of required connectivity steps. Statistical comparisons between groups using Student's *t*-tests across these four levels emphasize faster SFC to sensory, dorsal attention, and salience networks in ASD compared to controls, but delayed SFC to the DMN

(Supplementary Figures 8–10). Furthermore, SFC patterns and group differences were consistent when slightly changing the initial seed regions to coordinates in the vicinity of the sensory seeds published in earlier work[21]; notably, findings were even similar when seeding from intermediary, (non-DMN) networks (Supplementary Figures 11–12). Finally, the replication dataset showed virtually identical results (Supplementary Figure 7), confirming delayed SFC to the DMN but early acceleration in low-level networks in ASD.

To integrate gradient and SFC findings, we presented SFC trajectories in a coordinate system spanned by the first two gradients (Fig. 3), a synoptic representation of the connectome hierarchy as suggested in the previous studies[8,17]. In controls, SFC consistently evolved from sensory and somatomotor networks situated at the bottom poles towards DMN core nodes at the top, following an almost straight line (Fig. 3c left). Conversely, and despite an overall contracted arrangement in ASD (i.e., lower principal gradient locations of the DMN, higher locations of sensory/sensorimotor networks), SFC transitions never fully converged at the top DMN regions (Fig. 3c right). Instead, they showed an initial acceleration of SFC at the unimodal regions, followed by delayed transitions,

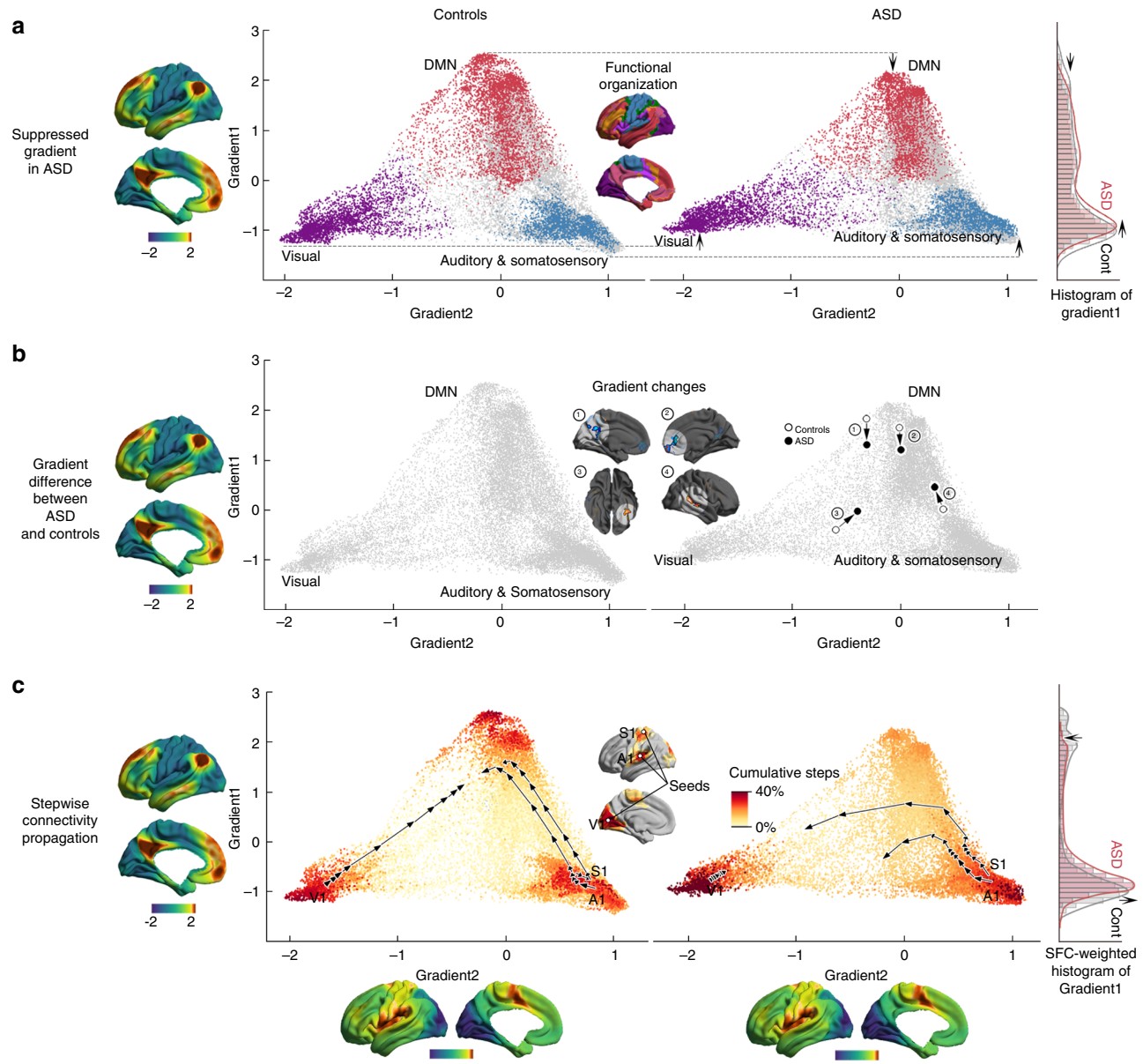

**Fig. 3** Integration of gradient and SFC findings. **a** Scatter plot of the first two connectivity embedding gradients in controls and ASD. Gradient 1 (*y*-axis) runs from primary sensorimotor (*dark turquoise*) to transmodal DMN (*sienna*). Gradient 2 (*x*-axis) separates somatomotor and auditory cortex from visual cortex. Triangular scattered points are colored with respect to established functional communities[18]. Histograms on right show the point density in ASD (light red) and controls (gray), suggesting overall compression of the first gradient in ASD. **b** Post-hoc analysis, showing positional shifts of the four significant clusters from the surface-based analysis (See Figure 1; 1-PCC/PCU, 2-mPFC, 3-OT, 4-pMTG). **c** Stepwise functional connectivity (SFC) in gradient space. Points are colored with respect to cumulative steps when simultaneously seeding from V1, S1, and A1. Trajectories (sampled every 20th step) illustrate the direct SFC from the primary sensory seeds to transmodal DMN in controls (left). ASD show an initially more rapid transition; however, trajectories deflect from a straight path and do not reach the DMN, even after 200 steps. Histogram on the right show point densities, weighted by the cumulative SFC. See also Supplementary Movies 1–3

culminating in slower and incomplete SFC to the DMN (Supplementary Movie 1–3).

**Relation to rich-club topology and connectivity distance.** Our next analysis stratified the gradient findings in the context of rich-club topology and connectivity distance. The former may play an important role in the functional communication of brain networks, where a core of high degree nodes with disproportionally strong interconnectivity is segregated from a more locally connected periphery. Specifically, rich-club nodes aggregate most long-range connections, while local/feeder nodes in the periphery show shorter connections[9]. To relate our findings to rich-club taxonomy and connectivity distance, we compared the gradient scores between ASD and controls across regions classified into either rich-club, feeder, or local nodes. This analysis revealed lower gradient scores in rich-club and feeder nodes in ASD, while local nodes showed increases, suggesting diminished segregation between the rich-club core and its periphery (Fig. 4a).

Furthermore, surface-wide correlation analysis between gradient group differences (i.e., the t-statistics from the between-group comparison, Fig. 1b) and connectivity distance indicated a positive relationship (Pearson correlation: $r = 0.6$, $p < 0.0001$), supporting preferential reductions in gradient values in regions with longer distance, while gradient values in primary regions with a short connectivity distance increased. When aggregating rich-club parameters and anatomical distance along SFC trajectories (Fig. 4b), we found that in controls, SFC initially ran along local/feeder nodes with short distances, but distance gradually increased after around 100 steps, co-occurring with a shift of transmodal rich-club activation. Notably, ASD cohort did not show selective rich-club convergence, and consequently no increase in overall connectivity distance, even after 200 steps.

**Relation to symptom severity.** Finally, we utilized supervised statistical learning to test whether network hierarchy features can provide a common explanation for both high and lower level symptoms in ASD. Using 5-fold cross-validations (where the classifier is repeatedly trained on 4 folds of the data and tested on the 5th fold), we found gradient and SFC features to significantly predict total ADOS total scores (mean average error, MAE = 2.42, median $r = 0.43$; permutation-test on performance exceeding chance levels $p < 0.001$) as well as ADOS subscores for social cognition (MAE = 1.91, median $r = 0.31$; $p < 0.006$) and repetitive behaviors/interest (MAE = 1.07, $r = 0.20$; permutation-test $p < 0.04$). Prediction performance was not significant for the communication subscale (permutation-test $p > 0.3$). Selected features differed across classifications, but most consistently included transmodal DMN (43%) and primary sensory areas (Visual: 17%, Sensory-motor: 10%) (Fig. 5). Consistent with prior findings suggesting that cross-validated accuracy is affected by sample size[32], classifier accuracy decreased but was still above chance for total ADOS and social cognition subdomain scores when predicting total ADOS using a leave-one-site-out strategy instead of 5-fold cross-validation (MAE = 2.09/2.26, median $r = 0.29/0.30$, permutation-test $p < 0.003$). Prediction for communication and repeated behavior did not reach significance (permutation-test $p > 0.1$).

**Motion effects.** Head motion can artificially change rs-fMRI connectivity measures, and thus we conducted three additional analyses to address its effects. Firstly, we repeated between-group comparisons after including mean framewise displacement as additional control covariate[33]. Secondly, we selected the 50% of individuals with ASD who had the lowest head motion and a motion-matched subgroup of 50% controls. Using this motion-minimized cohort, we carried out two analyses: (1) taking the gradient score from those original significant clusters and evaluating whether ASD still showed a meaningful group difference effect size compared to healthy controls, and (2) carrying out the SFC analysis and assessing if our original findings were replicated from this low-motion data. Finally, we correlated the individual z-scores of SFC anomalies with average framewise displacement, to assess whether degree of anomalies correlated with head motion estimates. None of these verification experiments indicated a noticeable impact of head motion. See Supplementary Figures 13–15 for details.

**Discussion**

Together with its parallel and modular architecture, network hierarchy has been widely recognized as a key principle of human brain organization. It has been observed across multiple subsystems, including sensory, motor, and higher-order transmodal networks[8,34]. Hierarchies are thought to guide the flow of information across the cortex, allowing sensory signals to become increasingly bound to other information and transformed into more abstract representations[35]. This architecture also imposes a natural separation between sensory-motor interactions with the outside world as well as existing representations that support complex inferences, but that need to be differentiated from external inputs, for example when interacting with other people[10]. Since ASD has been described as a disorder of information processing across multiple functional domains[36], here we examined the pattern of cortical functional hierarchy in two large samples of ASD and neurotypical controls, based on a novel combination of advanced connectome analytics. Indeed, our connectome gradient mapping showed that both groups revealed an axis of connectivity variations with low-level sensory systems and transmodal DMN on two opposite ends and remaining networks in-between. However, in ASD this gradient was globally contracted compared to the neurotypical group. Consistent with these results, the complementary analysis of SFC also demonstrated that in ASD sensory-driven connectivity transition do not converge to transmodal areas. Further graph theoretical stratification of the results revealed hierarchical imbalances related to a failure to selectively activate the rich-club core, suggesting the reduced capacity of the network backbone to switch into a global and long-range network communication pattern. Collectively, our multi-method approach provides converging evidence for atypical connectome hierarchy as a system-level substrate of ASD, a finding that could be replicated across distinct datasets, parameter choices, and when controlling for motion and various methodological confounds.

The present results suggest a diminished segregation in ASD between sensory systems and unimodal convergence regions such as pMTG and OT[37,38] on the one hand, and transmodal hubs such as mPFC and PCC/PCU on the other hand. In typically developing individuals, an important role of OT and pMTG in the integration of feedforward and feedback streams has previously been recognized in the visual and auditory system, particularly for face[37] and language processing[39]. Consistent with the clinical impairment in these domains, ASD-related atypical activations of these regions have been highlighted in several task-based fMRI studies[40]. DMN core nodes, such as mPFC and PCC/PCU, are among the most consistently activated regions during self-referential and introspective cognition[41] as well as other-oriented cognitive perspective-taking operations including mentalizing[42]. Again, these processes are highly impaired in ASD[43]. Across these and other cortical nodes, a series of previous rs-fMRI studies have already reported a series of results indicative of functional connectivity alterations in ASD[44]. However, findings

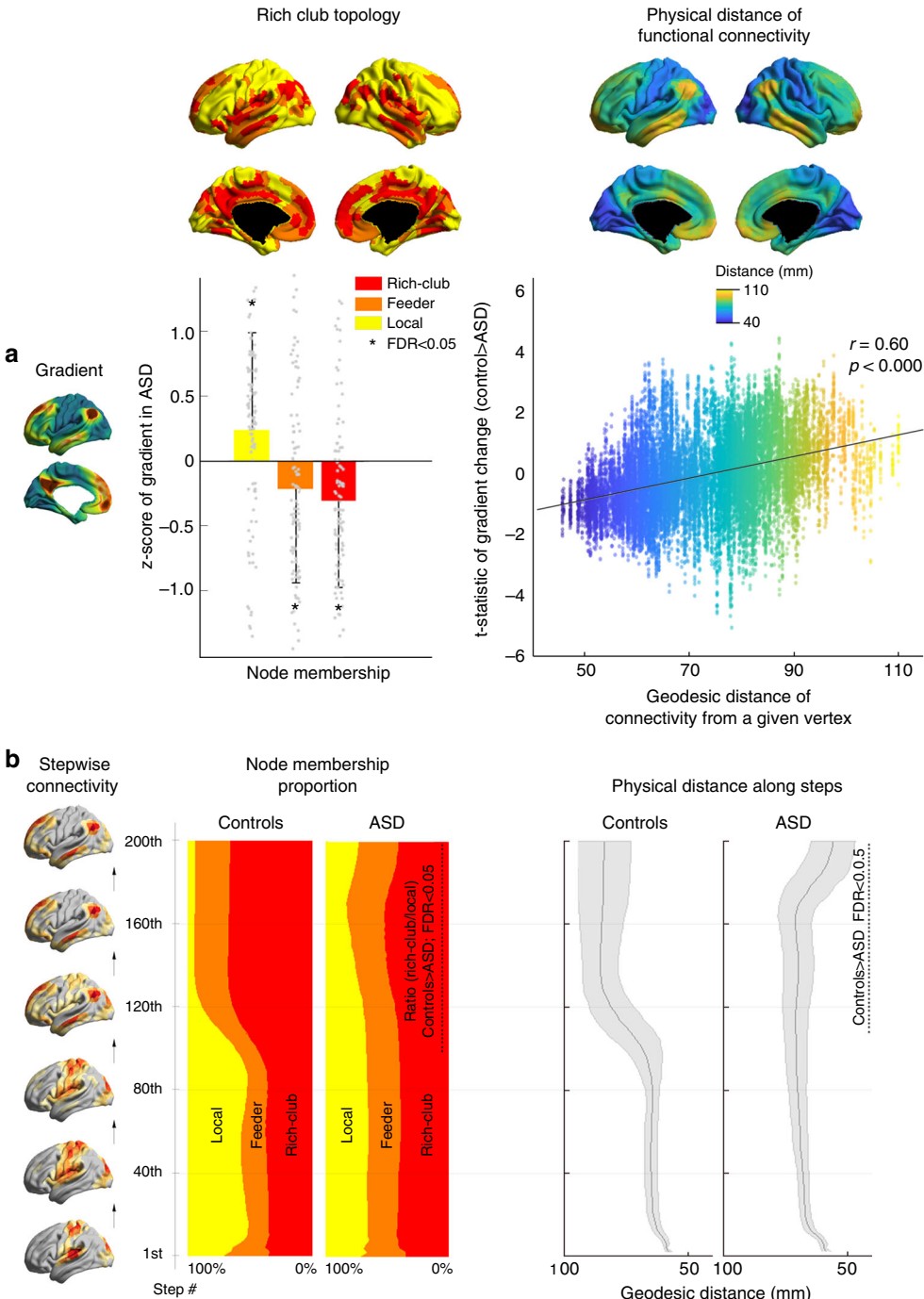

**Fig. 4** Relationship to rich-club and physical distance of functional connections. The top left **a** illustrates the results of the gradient comparisons and rich-club taxonomy, showing gradient reductions in feeder and rich-club nodes in ASD compared to controls (based on Student's *t*-tests) but increases in local nodes. The box plot indicates the mean ± SD. The top right **a** shows the vertex-wise association of between-group differences in gradient scores (t-statistic of the map from Fig. 1b; shown on the *y*-axis) and the average geodesic distance of functional connections of a given vertex (*x*-axis). The positive association indicates that regions with long-range connections tend to show gradient reductions (Pearson correlation coefficient *r* = 0.6). The bottom left **b** panel shows the proportion of visited nodes as a function of SFC steps in controls and ASD, illustrating the lower proportion of active rich-club vs local nodes from 100–200 steps. The bottom right **b** shows significantly shorter average connectivity distances in autism at these higher steps, established using Student's *t*-tests with multiple comparisons correction

appear heterogeneous in location and direction. Indeed, earlier studies with moderate sample size predominantly focusing on regions in the DMN have largely reported underconnectivity in ASD relative to controls[45,46], while others reported some exceptions[47,48]. These heterogeneous results can be in part attributed to

diversity in study parameters, participant inclusion criteria, imaging acquisition, processing, and data quality control[49,50]. However, more recent studies focusing on macroscale networks in larger samples have reconciled seemingly discordant findings of disconnection by revealing the co-occurrence of ASD-related

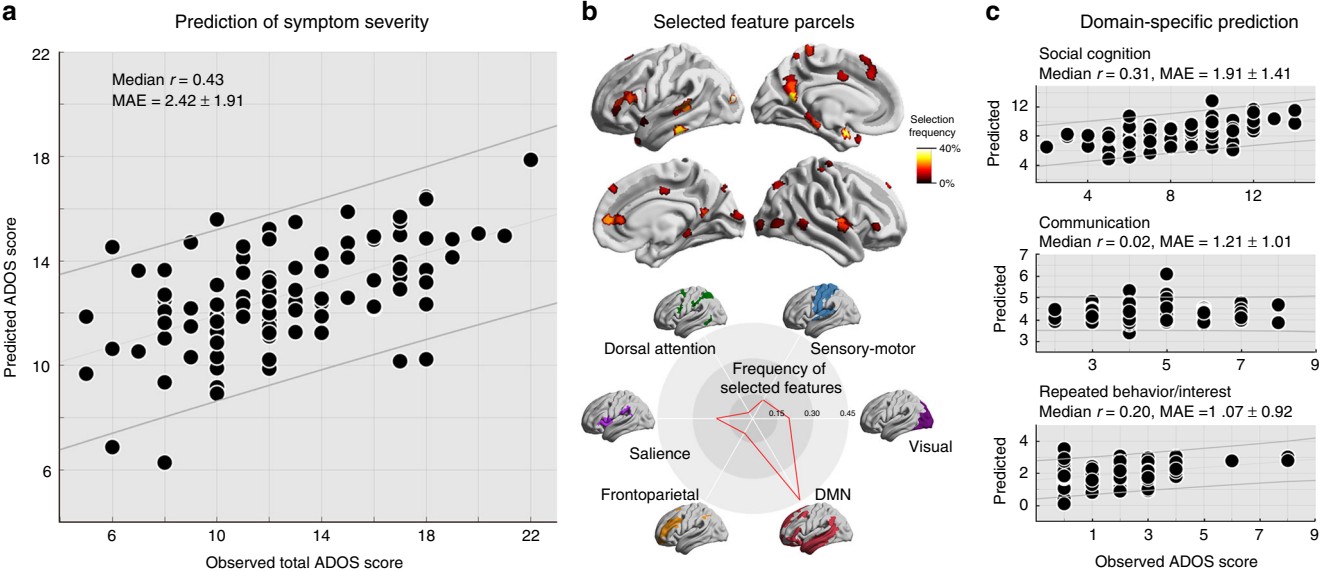

**Fig. 5** Relation to ASD symptom severity. **a** A supervised learning algorithm with 5-fold cross-validation was trained on gradient and SFC features to predict total ADOS scores and subscores in individuals with ASD. Significant accuracy was achieved for total ADOS scale (*shown*) and subscales for social cognition and repetitive behavior/interest (permutation-tests; *p* < 0.05). **b** Selected features across 100 iterations of the cross-validation, showing frequently selected features in hot colors. Community-based stratification[18] illustrates that most features were selected in visual, salience, and the DMN. **c** Prediction of subdomains (i.e., social cognition, communication, repetitive behavior/interest)

over-connectivity and under-connectivity ([51] and[44] for reviews). This mosaic pattern appears to be specific to the functional circuit employed, with cortico-cortical networks largely described as hypo-connected in ASD in accordance with the current findings, while subcortico-cortical circuits appear hyperconnected, particularly between the thalamus and somatomotor cortex[52,53]. In parallel, few recent studies supported the emerging concept that intrinsic functional connectivity networks in ASD may be idiosyncratically organized[54,55], where ASD expresses greater variability in functional network organization compared to controls, with stronger effects in default mode and somatomotor connectivity[56]. Collectively, these findings suggest that a broader ASD phenotype might arise from imbalances in functional architecture across individuals that are not purely anchored on spatial constraints but rather network-level features. Our proposed reference frame based on gradual connectome transitions and stepwise connectivity analyses targeted such overarching principles that could explain diverse behavioral phenotypes of ASD in a unified framework, as opposed to interrogating specific functional circuits. Notably, this approach provides a complementary viewpoint to conventional parcellation-based studies of macroscale brain organization and connectivity, and sidesteps the need to define discrete networks, such as those obtained through clustering techniques[18] or through recently developed boundary mapping techniques that delineate rapid transitions between connectivity patterns of different cortical points[19,20]. Although our method also leverages information of connectional affinity between areas similar to previous parcellation studies, the gradient mapping further projects these into a non-linear diffusion space and identifies principal components that describe main spatial axes in connectivity variations at the cortex-wide level. Therefore, the resultant gradient scores do not simply recapitulate network correlations/anticorrelations, as seen in their patterns of the DMN that is relatively close to some "task-positive" systems (e.g., the frontoparietal network), while it is further away from others, such as the sensory/motor cortex.

In addition to offering a novel perspective on connectional anomalies in autism, we incorporated a more conventional graph-theoretical rich club taxonomy as well as spatial geodesic distance into our analytical stream. Using this approach, we found that rich club nodes showed the strongest reduction in gradient scores and observed more marked reductions in areas with long-range connectivity compared to those with short-range functional connectivity profiles. Notably, our study derived a rich-club backbone from both controls and ASD to stratify differences in gradient and SFC analyses, while other previous studies on rich club architecture first identified this subnetwork in ASD and controls separately, then compared their overall connectivity and spatial configuration between the groups[57,58]. As such, a direct link of our findings and those in previous studies may not be straightforward. Yet, our findings are consistent with other reports of decreased network centrality, an alternative measure of node influence, across multiple cortical regions in individuals with ASD[59,60]. Overall our results are in parallel to those previous studies that ASD is characterized by spatial and topology-level reorganization of this core subnetwork. Thus, beyond recapitulating previous and seemingly fragmented findings including reduced cortico-cortical functional connectivity between transmodal and sensorimotor networks[52,61], our approach provides more direct measures of atypical connectome cortical hierarchy as a system-level substrate of ASD. The availability of such objective markers will allow for further examinations across disorders to identify specific and transdiagnostic atypicalities, an effort that is becoming closer to reach with the sharing of transdiagnostic samples, such as the healthy brain network (https://healthybrainnetwork.org).

An additional appealing feature of our functional hierarchy perspective on ASD is that it may be relevant for future work targeting brain anomalies at macroscopic, microcircuit, and molecular scales. Studies in early brain development have suggested atypical brain growth and cortical maturation in autism[62], which may potentially disrupt the spatial arrangement of functional networks and thus have a measurable effect on network

topographies and the hierarchical embedding of different brain regions[22,63]. At the microcircuit level, atypical hierarchical organization may be related to anomalies in local signaling, specific perturbations in the balance of excitation and inhibition (E/I). In Shank3 mice, an ASD model with established E/I imbalance related to mutations in synaptic scaffolding, recent work has shown reduced prefrontal functional connectivity to other higher order regions that were found to be predictive of intellectual disability and socio-communicative impairments[64]. Similar effects were demonstrated a different model related to mutations in the cell adhesion molecule CNTNAP2, where reductions in long-range rs-fMRI connectivity affected heteromodal fronto-posterior components of the mouse DMN, an effect that was associated with reduced social investigation[65]. Other molecular studies have also highlighted genes with effects on GABA/Glutamate pathways in ASD[66] and associated to the broader behavioral phenotype that includes both sensory anomalies, as well as atypical social interactions[67]. These studies reinforce the concept that such imbalances may serve as a pathophysiological mechanism of ASD[6]. While in-vivo work on E/I imbalance is scarce in the ASD field, recent work in other neuropsychiatric conditions has used computational analyses utilizing information of brain network hierarchy to assess consequences of physiological imbalance, ultimately providing a framework to bridge macrolevel anomalies and microcircuit properties[68]. In this way our focus on network hierarchies may represent a first step towards the development of novel computational accounts that can enrich our understanding of system-level dysfunction in autism, extending previous imaging work that has so far mainly focused on mapping morphological anomalies in specific areas or the study of selected functional connections[69].

These findings of atypical network hierarchy offer a novel and parsimonious account of the range of symptoms observed in ASD that encompass multiple domains across sensory-motor, cognitive and social-communicative functioning. Indeed, gradient and SFC features could guide a supervised learning algorithm to predict symptom severity in individuals with ASD. Predictive features were observed in the DMN, as well as sensory systems, indicating that both sensory and higher-order aspects of the cortical hierarchy underpin ASD symptomatology. Notably, however, a causal direction between low- and high-level network anomalies cannot be established from our cross-sectional design; yet post-mortem studies suggest differences in developmental-maturational trajectories across brain regions that recapitulate macroscale functional gradients, with sensory areas showing early myelination and maturation, while transmodal centers show a more protracted development[70]. Recently, it has been also established that the transmodal end of the functional gradient could support detailed experiences based on visual input[71]. Given this evidence, early perturbations in the development of unimodal regions may ultimately express a cascading effect, affecting higher-order networks at each subsequent stage of the hierarchy. This "sensory-first hypothesis"[6] also draws support from experimental studies demonstrating hypersensitivity to sensory stimuli in the early years, which may predict subsequent impairments in social interaction in ASD[72]. Interestingly, separately analyzing children and adults in our dataset revealed disruptions in early systems in both, while DMN anomalies were more marked in adults with the condition. Please note that, however, the cross-sectional nature of our sample essentially prevents us from specifying true developmental roles of sensory and transmodal system atypicality in ASD. Future work, therefore, requires to be conducted based on more systemically controlled samples such as a longitudinal data of individuals with autism, often underrepresented from early childhood[24].

We close by noting that our findings of dysfunctional macro-scale integration and segregation in ASD was made possible by the open ABIDE data repository. These initiatives offer the neuroimaging and network neuroscience communities an unprecedented access to large datasets for the investigation of healthy and diseased brains. As they have also highlighted variability in data quality across sites, our work involved an extensive case screening, several statistical corrections, and validation experiments to confirm the robustness of our findings. Specifically, we verified our results in variations of MRI quality and motion, given that such confounds have been suggested to contribute to the marked heterogeneity in previous results in ASD[15]. In contrast to homogenized multisite data, heterogeneous datasets such as ABIDE may pose unique constraints on the accuracy of predictive routines that operate from one site to another. Indeed, a leave-one-site-out learner did not achieve as high accuracy as a 5-fold cross-validation learner that pooled data from sites. Notably, the employed 5-fold cross-validation nevertheless provided a more thorough control on classifier accuracy compared to commonly applied within-sample regression or leave-one-out cross-validation. We could also observe similar main effects across included sites in the discovery dataset and replicated findings in an independent verification cohort from the second ABIDE wave. Given that reproducibility is increasingly important, not only in the context of psychiatric research, our study shows that the advantages to our understanding of complex diseases such as ASD made possible by open-access initiatives outweigh any costs associated with data quality.

## Methods

**Participants.** We studied two datasets (i.e., discovery and replication) selected from each wave of the openly-shared Autism Brain Imaging Data Exchange initiative (ABIDEI and II; http://fcon_1000.projects.nitrc.org/indi/abide)[23,24].

The discovery data originated from a previously described ABIDE-I subsample[23,24]. Briefly, we selected those sites that included children and adults with ASD and typical controls, with ≥10 individuals/group (n = 287, 143 ASD, 144 healthy controls). We restricted the analysis to males given the low prevalence of females in ABIDE-I. Detailed quality control included only cases with acceptable T1-weighted MRI, surface-extraction and head motion in rs-fMRI (see below), resulting in 211 individuals from three sites: (1) NYU Langone Medical Center (NYU, 35/51 ASD/controls); (2) University of Utah, School of Medicine (USM, 49/37 ASD/controls); (3) University of Pittsburg, School of Medicine (PITT, 19/20 ASD/controls).

The replication data was a subsample of ABIDE-II[24]. Inclusion criteria were similar as for the Discovery data, but also included females given an increased female ratio in ABIDE-II. We initially selected 177 individuals from three sites. After quality control as for the Discovery dataset, we included 103 individuals: (1) Trinity Centre for Health Sciences, Trinity College Dublin (TCD, 12/16 ASD/controls); (2) NYU Langone Medical Center (NYU, 25/18 ASD/controls); (3) Institut Pasteur/Robert Debré Hospital (IP, 11/21 ASD/controls).

Individuals with ASD underwent structured or unstructured in-person interview and had diagnosis of Autistic, Asperger's, or Pervasive Developmental Disorder Not-Otherwise-Specified established by expert clinical opinion aided by 'gold standard' diagnostics: Autism Diagnostic Observation Schedule, ADOS[25] and/or Autism Diagnostic Interview-Revised, ADI-R. These focus on three domains including reciprocal social interactions, communication and language, and restricted/repeated behaviors. Full scale/performance/verbal IQ was measured via WASI, WAIS III, and/or WISC III. Controls had no history of mental disorders and were statistically matched for age to the ASD group at each site. In the ABIDE-I and –II datasets, there were no differences in age and sex between controls and ASD. ABIDE-I and -II datasets are based on studies approved by local IRBs, and data were fully anonymized (removing all HIPAA protected health information identifiers, and face information from structural images).

**MRI acquisition.** High-resolution T1-weighted images (T1w) and resting-state functional MRIs (rs-fMRI) were available from all sites and in both discovery and replication datasets. Images were acquired on 3 T scanners from Siemens (NYU, USM, PITT) or Philips (IP, TCD). NYU data were acquired on an Allegra using 3D-TurboFLASH for T1w (TR = 2530 ms; TE = 3.25 ms; TI = 1100 ms; flip angle = 7°; matrix = 256 × 256; 1.3 × 1.0 × 1.3 mm3 voxels) and 2D-EPI for rs-fMRI (TR = 2000ms; TE = 15 ms; flip angle = 90°; matrix = 80 × 80; 180 volumes, 3.0 × 3.0 × 4.0 mm3 voxels). PITT data were acquired on an Allegra using 3D-MPRAGE for T1w (TR = 2100 ms; TE = 3.93 ms; TI = 1000 ms; flip angle = 7°; matrix = 269 × 269;

$1.1 \times 1.1 \times 1.1$ mm$^3$ voxels) and 2D-EPI for rs-fMRI (TR = 1500 ms; TE = 35 ms; flip angle = 70°; matrix = 64 × 64; 200 volumes, $3.1 \times 3.1 \times 4.0$ mm$^3$ voxels). *USM* data were acquired on a TrioTim using 3D-MPRAGE for T1w (TR = 2300 ms; TE = 2.91 ms; TI = 900 ms; flip angle = 9°; matrix = 240 × 256; $1.0 \times 1.0 \times 1.2$ mm$^3$ voxels) and 2D-EPI for rs-fMRI (TR = 2000ms; TE = 28 ms; flip angle = 90°; matrix = 64 × 64; 240 volumes; $3.4 \times 3.4 \times 3.0$ mm$^3$ voxels). *TCD* data were acquired on an Achieva using 3D-MPRAGE for T1w (TR = 3000 ms; TE = 3.90 ms; TI = 1150 ms; flip angle = 8°; matrix = 256 × 256; $0.9 \times 0.9 \times 0.9$ mm$^3$ voxels) and 2D-EPI for rs-fMRI (TR = 2000ms; TE = 27 ms; flip angle = 90°; matrix = 80 × 80; 210 volumes; $3.0 \times 3.0 \times 3.2$ mm$^3$ voxels). *IP* data were acquired on an Achieva using 3D-MPRAGE for T1w (TR = 2500 ms; TE = 5.60 ms; flip angle = 30°; matrix = 240 × 240; $1 \times 1 \times 1$ mm$^3$ voxels) and 2D-EPI for rs-fMRI (TR = 2700 ms; TE = 45 ms; flip angle = 90°; matrix = 64 × 63; 85 volumes; $3.59 \times 3.65 \times 4$ mm$^3$ voxels).

**MRI processing**. The following processing was applied to discovery and replication samples.

Structural MRI processing was based on FreeSurfer (v5.1; http://surfer.nmr. mgh.harvard.edu/). Image processing included bias field correction, registration to stereotaxic space, intensity normalization, skull-stripping, and white matter segmentation. A triangular surface tessellation fitted a deformable mesh model onto the white matter volume, providing gray-white and pial surfaces with >160,000 corresponding vertices. Individual surfaces were registered to an average template surface with a spherical representation, fsaverage, improving correspondence of measurement points with regards to sulco-gyral patterns.

Our rs-fMRI analysis was based on preprocessed data previously made available by the Preprocessed Connectomes initiative (http://preprocessed-connectomes-project.org/abide/). Processing was based on C-PAC (https://fcp-indi.github.io/) and included slice-time correction, head motion correction, skull stripping, and intensity normalization. Statistical corrections removed effects of head motion, white matter and cerebrospinal fluid signals using the CompCor tool, based on the top 5 principal components[73], as well as linear/quadratic trends. After band-pass filtering (0.01–0.1 Hz), we co-registered rs-fMRI and T1w data in MNI152 space through combined linear and non-linear transformations. Surface alignment was verified for each case and we interpolated voxel-wise rs-fMRI time-series along the mid-thickness surface. We resampled rs-fMRI surface data to Conte69, a template mesh from the Human Connectome Project pipeline (https://github.com/Washington-University/Pipelines) and applied surface-based smoothing (FWHM = 5 mm). MRI quality control was complemented by assessment of signal-to-noise ratio and visual scoring of surface extractions for structural MRI, and evaluation of temporal derivatives and framewise displacement for rs-fMRI[33] (Supplementary Figure 1).

Cortical surface extraction was visually inspected. Subjects with severe faulty segmentations, other artifacts or more than 0.3 mm framewise displacement were excluded (discovery: n = 76/287, replication: n = 74/177). Segmentation inaccuracies of all remaining cases were manually corrected.

**Connectome gradient analysis**. Surfaces and their features were downsampled to 10k vertices per hemisphere for computational efficiency. We generated cortex-wide connectome gradients for each participant based on open software (https://github.com/NeuroanatomyAndConnectivity/gradient_analysis). Using the rs-fMRI time-series matrix in each subject, we calculated functional connectome based on systematic Pearson correlations (with 20,484 × 20,484 entries). As in a previous study[17], we z-transformed and thresholded this matrix, leaving only the top 10% of weighted connections per row, and calculated a cosine similarity matrix that captures similarity in connectivity profiles between vertices. We applied diffusion map embedding[17,26], a nonlinear manifold learning approach, to identify principal gradient components explaining connectome variance in descending order (each of $1 \times 20,484$). In brief, the algorithm estimates a low-dimensional embedding from a high-dimensional connectome matrix. In this space, cortical vertices that are strongly interconnected by either many connections or few very strong connections are closer together, whereas vertices with only little or no inter-connectivity are farther apart. The name of this approach, which belongs to the family of graph Laplacians, derives from the equivalence of the Euclidean distance between points in the embedded space and the diffusion distance between probability distributions centered at those points. Compared to other non-linear manifold learning techniques, the diffusion maps algorithm is relatively robust to noise and computationally inexpensive. Notably, the algorithm is controlled by a single parameter α, which controls the influence of density of sampling points on the manifold (α = 0, maximal influence; α = 1, no influence). In this study, we followed previous recommendation and set α = 0.5, a choice that retains the global relations between data points in the embedded space. Following estimation of a group-level gradient component template, generated from an average connectivity matrix based on both ASD and controls, we performed Procrustes rotation to align components of each individual to the template[27]. As a previous study in healthy individuals[17], we focused on the first two components which explained 37% variance (Supplementary Figure 2 also shows components 3–20). Components explained similar variance in ASD and controls (i.e., no statistical difference; two-tailed Student's t-test; $t < 1.30$, $p > 0.1$). These components, initially defined in connectivity space, were then mapped back onto the cortical

surface to visualize macroscale transitions in overall connectivity patterns at the system-level.

Gradient component scores were statistically compared between ASD and controls using surface-based linear models implemented in SurfStat (http://www.math.mcgill.ca/keith/surfstat/) for Matlab. Models included effects of age, site, and group. Surface-based findings were corrected for family-wise errors due to multiple comparisons using a random field theory of $p_{FWE} < 0.05$.

**Stepwise connectivity estimation**. Stepwise functional connectivity (SFC) extends conventional seed-based connectivity analyses by detecting not only direct inter-regional but also indirect—yet meaningful—associations along successive connectivity steps. It counts the number of all paths that connect a given cortical region/vertex to the target region/vertex in a given length of connectivity distance. In other words, at a step-distance $l$, the degree of SFC at a target node $i$ was computed by counting all possible paths between $i$ and the seed that has an exact length of $l$. In our study, SFC was generated for each subject using the *findwalks.m* function from the Brain Connectivity Toolbox. We used the same thresholded matrix as input in the gradient mapping analysis. As the degree of SFC exponentially grows with increasing steps, we standardized the value at each step by subtracting the whole brain mean of a SFC map and dividing by its standard deviation[74]. Therefore, each SFC map represents a relative increase of connectivity degree across different step-distances. It has been shown that sensory-driven SFC maps in healthy individuals reach the transmodal DMN core after a certain number of steps, suggesting that the DMN may be at the top of a functional hierarchy[17]. For an integrative analysis, we put seed areas simultaneously in three representative primary sensory cortices, including visual (V1; MNI coordinate x, y, z: −14/10 [left/right], −78, 8), auditory (A1; −54/58, −14, 8) and somatosensory (S1; −42/38, −29, 65) areas. Notably to build a robust SFC degree map at the individual level, we first estimated a group-level template for the ASD and control groups. In the next step, we build individual SFC maps based on subject-specific connectomes but also guided by group-level SFC, by multiplying it with a subject SFC map at each step. This two-step group-guided approach was chosen to mitigate recursive SFC calculations that were observed at the individual level. In controls, SFC maps generally converged in the DMN after $l = 200$ steps. In addition to visualizing SFC maps in ASD and controls, we estimated how many SFC steps were required to reach a certain level of hierarchy, and compared between groups using t-tests (two-side) with multiple comparisons correction. This calculation was proceeded by assigning functional communities[17] to levels of hierarchy from a model initially proposed by Mesulam (Fig. 2; level 1: sensory and somatomotor networks, level 2: dorsal attention, salience networks, level 3: frontoparietal network, level 4: DMN).

**Relation to rich-club topology**. We used functions from the Brain Connectivity Toolbox (https://sites.google.com/site/bctnet/) to identify the rich-club. For computational efficiency, we subdivided the cortex into 1004 similarly-sized parcels[75], computed a weighted functional connectome (1004 × 1004), and applied the same thresholding as in the above analyses. We computed a rich-club coefficient $\varphi(k)$ per individual, varying $k$ from 1 to the maximal degree. We normalized $\varphi(k)$ relative to 1000 randomly rewired networks, resulting in $\varphi_{norm}(k)$ and determined the $k'$ that maximized $\varphi_{norm}(k)$ in controls and ASD. We identified rich-club nodes with degree greater or equal to $k'$. Remaining nodes were classified as feeders (≥10% of connections to rich-club nodes) or local nodes (<10%). To assess the interplay between rich-club and gradient findings, we compared gradient scores across all rich-club, feeder, and local nodes between ASD and controls, using Student's t-tests with multiple comparison correction at a false discovery level of 0.05 ($p_{FDR} < 0.05$). To integrate SFC findings with rich-club hierarchy, we tracked the proportion of rich-club, feeder, and local nodes at every step, and computed a ratio of rich-club-vs-local nodes. This ratio was compared between groups using two-sample t-tests ($p_{FDR} < 0.05$).

**Relation to connectivity distance**. We calculated geodesic distances along the cortical mantle using the Fast Marching Toolbox between all pairs of surface points (https://github.com/gpeyre/matlab-toolboxes/tree/master/). We binarized the above thresholded functional matrix, taking only positive connectivity and multiplied it with the resulting distance matrix. We finally calculated row-wise sums of this matrix and divided it by the overall node degree, which conceptually provides the average distance of significant functional connections for a given region. To assess associations between gradient changes in autism and connectivity distance, we correlated the surface-based distance map with the t-statistical map of the between-group difference in the principle functional gradient. Furthermore, we tracked average connectional distance across all SFC steps.

**Symptom severity prediction**. We used supervised pattern learning to associate our imaging markers to ASD symptom severity. Symptom severity was indexed by ADOS total and sub-scores (i.e., social interaction, communication, repeated behaviors/interests). We tested the ABIDE-I dataset in this analysis since all participants in this sample had ADOS scores collected. Training and performance evaluation employed nested 5-fold cross-validation with 100 repetitions. The classifier operated on the first score of the principal gradient and four SFC maps

that were averaged across each level of functional hierarchy (see stepwise connectivity estimation, above, for details). As for the rich-club analysis, we averaged features using a 1004-ROI parcellation, yielding 5020 (1004 × 5) features for each individual. Cross-validation incorporated training and testing. Training: 4 folds were split into 3 internal training and 1 validation subfold. A feature selection procedure was carried out based using an elastic net (alpha = 0.5, to weight LASSO [least absolute shrinkage and selection operator] and ridge regression equally), which generated at every iteration a subset of imaging features that correlate to ADOS scores in the training dataset. Selected features trained the main classifier, for which support vector regression was chosen[32]. 4-fold internal cross-validations effectively trained the main classifier to optimally predict ADOS scores during validation. Testing: The remaining unseen 'test' fold was used to calculate prediction accuracy. We repeated the training-test procedure across all 5 folds for 100 times, and calculated median performance. Observed/predicted ADOS scores were compared using Pearson correlation coefficients, $r$, and mean absolute error, MAE. A total of 1000 permutation tests with randomly shuffled ADOS scores determined whether our prediction (at median performance) exceeded chance level. In a separate analysis, we evaluated predictive performance when using a leave-one-site-out strategy, where we trained the classifier on data from two sites and tested it on data from the remaining site. As the feature selection procedure provided slightly different features in each iteration (depending on the internal data folds in the training phase), we repeated the experiment 100 times and calculated average performance. Again, we corrected the features for age and site effects prior to the prediction experiments.

**Robustness analyses**. To demonstrate robustness of our main gradient and SFC findings with respect to several methodological choices, we performed several replication analyses.

- Given ongoing discussions on the effects of Global Signal Regression (GSR) on rs-fMRI data[30], we did not perform GSR for our main analyses. We nevertheless repeated gradient and SFC analyses using GSR-processed data from Preprocessed Connectomes repository (http://preprocessed-connectomes-project.org/abide/).
- A recent study suggested that unspecific differences in overall connectivity strength may affect case-control network analyses[31]. We thus repeated our main analyses after additionally correcting for average whole-brain connectivity.
- To assess the effects of different matrix thresholding on our results, we furthermore repeated the gradient and SFC analyses at 5–25% thresholded matrices.
- SFC analyses were based on previously published coordinates[21]. To dispel an influence of the exact seed location, we systematically varied seed coordinates. To first assess effects of slight variations in the seed coordinates, we identified immediate neighboring parcels in V1, S1, and A1 (within a radius of 3 mm; black dots in Supplementary Figure 11) and randomly selected three seeds from those neighbors, each from either V1 or S1 or A1, and conducted a SFC analysis for reproducibility. We repeated this procedure 10 times. To evaluate SFC initialized from non-default mode networks, we also placed seeds in frontoparietal, salience, and dorsal attention networks (Supplementary Figure 12).
- We repeated surface-wide comparison of gradient scores using non-parametric permutation tests[28]. Briefly, we randomly shuffled group membership (i.e., ASD, controls) for each case and performed a cortex-wide comparison (cluster defining threshold α = 0.01). Repeating this procedure 10000 times, we aggregated the maximum cluster size as an empirical null distribution. We estimated the significance of our findings by placing the cluster sizes of the actual between-group comparison into the permutation distribution. As in the main analysis, we statistically corrected for age and site prior to any permutation.

**Reproducibility analysis**. Studying the replication dataset, we compared the principal gradient scores and SFC between ASD and controls. As our ABIDE-II sub-selection also included women, sex was additionally entered into all statistical models.

**Ethics**. In accordance with HIPAA guidelines and 1000 Functional Connectomes Project / INDI protocols, all ABIDE datasets have been anonymized, with no protected health information included.

## Data availability

All data analyzed in this manuscript were obtained from the open-access ABIDE-I and ABIDE-II initiatives, in either raw (http://fcon_1000.projects.nitrc.org/indi/abide/) or processed form (http://preprocessed-connectomes-project.org/abide/). Gradient mapping analyses was based on open-access tools (https://github.com/NeuroanatomyAndConnectivity/gradient_analysis). Stepwise functional connectivity and rich-club analyses made use of the brain connectivity toolbox (https://sites.google.com/

site/bctnet/home). Surface-wide statistical comparisons were carried out using SurfStat (http://www.math.mcgill.ca/keith/surfstat/).

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

## Acknowledgements

We would like to thank each site (http://fcon_1000.projects.nitrc.org/indi/abide/) and NITRC (http://www.nitrc.org) for providing the data sharing platform for the ABIDE initiative. Funding sources for the ABIDE-1 subsample: The *NYU* site was supported by NIH (K23MH087770; R21MH084126; R01MH081218; R01HD065282), Autism Speaks, The Stavros Niarchos Foundation, The Leon Levy Foundation, an endowment provided by Phyllis Green and Randolph Cowen. The *PITT* site was supported by Autism Speaks Grant 04593, KO1 NIMH MH081191, NIMH MH67924, NIH MH55748. National Institutes of Health (grant numbers: K08 MH092697, RO1MH080826, P50MH60450, T32DC008553, R01NS34783). The *USM* site was supported by Autism Speaks Mentor-based Predoctoral Fellowship (grant number: 1677), University of Utah Multidisciplinary Research Seed Grant, NRSA Predoctoral Fellowship (grant number: F31 DC010143), Ben B. and Iris M. Margolis Foundation. Reinder Vos de Wael was supported by the McGill Faculty of Medicine. Dr Hong was supported by a postdoc fellowship from the Canadian League Against Epilepsy as well as from CIHR. Dr Bethlehem acknowledges research support by the Autism Research Trust and a British Academy Fellowship (PF2\180017). Sara Larivière was supported by a Jeanne Timmins Costello Fellowship and FRQS. Dr Smallwood was supported by the European Research Council (WANDERINGMINDS-ERC646927). Dr Bernhardt acknowledges research support from NSERC (Discovery-1304413), CIHR (FDN-154298), Azrieli Center for Autism Research of the Montreal Neurological Institute (ACAR), SickKids Foundation (NI17–039), as well as salary support from FRQS (Chercheur

Boursier Junior 1). Dr Paquola was supported by a postdoctoral fellowship from the Transforming Autism Care Consortium (TACC). Drs Bethlehem and Bernhardt were furthermore supported by an MNI-Cambridge collaboration grant. Drs Di Martino and Milham were supported by NIMH (NIMH-105506 and NIMH-099059, respectively).

## Author contributions

S.J.H. and B.C.B. designed the study, carried out the image processing and data analysis, interpreted the findings, and wrote and revised the manuscript. J.S. assisted in the interpretation of findings and wrote and revised the manuscript. R.V.D.W. assisted in data analysis and revised the manuscript. R.A.I.B. and D.S.M. assisted in study design, interpretation of findings, and revised the manuscript. S.L., M.P.M., A.D., and C.P. assisted in interpretation of findings and revised the manuscript. S.L.V. assisted in image processing and cohort selection and revised the manuscript.

## Additional information

**Competing interests:** The authors declare no competing interests.

