## [Peer Review File · Nature Communications]

Editorial Note: This peer review file has been redacted to remove instances where text from additional reviewer comments (not part of the formal peer review) was quoted as we do not have permission to publish these.

Reviewers' comments:

Reviewer #1 (Remarks to the Author):

Hong and colleagues report functional connectivity differences between young adults with ASD vs. age-matched controls. The authors downloaded preprocessed resting state fMRI data from the Autism Brain Imaging Data Exchange (ABIDE) initiative, which contains a heterogeneous accumulation of inconsistently acquired data from multiple institutions. The current report is based on a particular mode of resting state fMRI analysis, first described by Margulies et al. (2016), in which "diffusion embedding" is used to effect dimensionality reduction of covariance structures in manner similar to principal component analysis (PCA). The major claim here is that the magnitude of "gradient scores" (effectively, principal components) in the ASD group is systematically weaker than that obtained in controls. As illustrated in Figure 1, this lower magnitude appears as a modest reduction in functional connectivity strength. These results are interpreted as demonstrating "perturbed propagation of intrinsic flow from sensory towards transmodal areas" which finding is related to the "deficits in social cognition and low-level behavioral symptoms, but not problems in communication" that are said [inaccurately regarding impaired communication] to be characteristic of ASD.

The first major problem with this work is total failure of appropriate scholarship. The literature cited by the authors primarily concerns the behavioral characteristics of ASD. Pointedly omitted is any mention of prior resting state fMRI studies comparing ASD to controls. A non-expert could easily be misled into assuming that this paper represents the first time that functional connectivity abnormalities have been associated with ASD. In fact, crossing "ASD" x "resting state fMRI" in PubMed currently returns 140 hits of which 7 are reviews. The history of resting state fMRI in ASD now becomes relevant. A substantial but massively contradictory literature on this topic had accumulated as of 2012 at which juncture it became clear that virtually all extant studies were corrupted by systematically greater head motion artifact in ASD (Deen and Pelphrey, 2012). Since then, most studies, including the present work, have taken care to control for this confounding factor.

However, a consistent pattern of functional connectivity changes associated with ASD has not emerged. Most recent studies have taken a data-driven approach in which classifiers of various design are used to identify discriminating functional connectivity features. Classifier accuracy ranges from 60% (Nielsen et al., 2013) or 67% (Abraham et al., 2017) to 91% (Chen et al., 2015), depending on the selectivity of the inclusion criteria. Clear focality is notably absent over-all. Thus, discriminating node pairs derived by unsupervised classification are widely distributed over the whole brain, e.g., (Heinsfeld et al., 2018; Plitt et al., 2015), possibility with an emphasis on more posterior regions, e.g., (Keown et al., 2013). Alternatively, multiple functional systems, e.g., "salience, default mode, frontotemporal, motor, and visual networks" are reported as affected (Uddin et al., 2013). Or, functional connectivity is said to be globally "idiosyncratic" (Nunes et al., 2018). Curiously, recent reports tend to emphasize over- as opposed to under-connectivity (Plitt et al., 2015; Yahata et al., 2016), i.e., precisely the inverse of what is here reported. This discrepancy is, of course, not discussed.

The second major problem is the credulous adoption of a particular approach to functional connectivity analysis as described by Margulies et al., 2016. Allowing that that work makes a worthwhile point concerning the representation of function within the cerebral cortex of normal subjects, it does not follow that "principal gradient" decomposition is uniquely suited to discovering how ASD alters resting state functional connectivity. But that is the implication, since all other approaches are ignored. It should be noted that, as visualized on the brain surface, the first few "principal gradients" closely resemble the functional connectivity components previously obtained by others using related but more straightforward methods, e.g., (Lee et al., 2012; Power et al., 2011; Yeo et al., 2011). Thus, if, the present results have any relevance to ASD, similar findings should be obtained by more straightforward methods. Absent such a demonstration, the reported

results simply reflect how ASD appears as viewed through a particular lens. Given the diffuse findings reported by others, including findings obtained using the same ABIDE dataset, it is predictable that "principal gradients" abnormalities would be found in ASD. But, it remains unclear that anything fundamental has been learned about ASD. In any case, the authors are obliged to reconcile their finding of weaker functional connectivity in ASD vis a vis much of the recent literature.

- Abraham, A., Milham, M.P., Di Martino, A., Craddock, R.C., Samaras, D., Thirion, B., Varoquaux, G., 2017. Deriving reproducible biomarkers from multi-site resting-state data: An Autism-based example. *Neuroimage* 147, 736-745.
- Chen, C.P., Keown, C.L., Jahedi, A., Nair, A., Pflieger, M.E., Bailey, B.A., Muller, R.A., 2015. Diagnostic classification of intrinsic functional connectivity highlights somatosensory, default mode, and visual regions in autism. *Neuroimage Clin* 8, 238-245.
- Deen, B., Pelphrey, K., 2012. Perspective: Brain scans need a rethink. *Nature* 491, S20.
- Heinsfeld, A.S., Franco, A.R., Craddock, R.C., Buchweitz, A., Meneguzzi, F., 2018. Identification of autism spectrum disorder using deep learning and the ABIDE dataset. *Neuroimage Clin* 17, 16-23.
- Keown, C.L., Shih, P., Nair, A., Peterson, N., Mulvey, M.E., Muller, R.A., 2013. Local functional overconnectivity in posterior brain regions is associated with symptom severity in autism spectrum disorders. *Cell Rep* 5, 567-572.
- Lee, M.H., Hacker, C.D., Snyder, A.Z., Corbetta, M., Zhang, D., Leuthardt, E.C., Shimony, J.S., 2012. Clustering of resting state networks. *PLoS One* 7, e40370.
- Nielsen, J.A., Zielinski, B.A., Fletcher, P.T., Alexander, A.L., Lange, N., Bigler, E.D., Lainhart, J.E., Anderson, J.S., 2013. Multisite functional connectivity MRI classification of autism: ABIDE results. *Front Hum Neurosci* 7, 599.
- Nunes, A.S., Peatfield, N., Vakorin, V., Doesburg, S.M., 2018. Idiosyncratic organization of cortical networks in autism spectrum disorder. *Neuroimage*.
- Plitt, M., Barnes, K.A., Martin, A., 2015. Functional connectivity classification of autism identifies highly predictive brain features but falls short of biomarker standards. *Neuroimage Clin* 7, 359-366.
- Power, J.D., Cohen, A.L., Nelson, S.M., Wig, G.S., Barnes, K.A., Church, J.A., Vogel, A.C., Laumann, T.O., Miezin, F.M., Schlaggar, B.L., Petersen, S.E., 2011. Functional network organization of the human brain. *Neuron* 72, 665-678.
- Uddin, L.Q., Supekar, K., Lynch, C.J., Khouzam, A., Phillips, J., Feinstein, C., Ryali, S., Menon, V., 2013. Salience network-based classification and prediction of symptom severity in children with autism. *JAMA Psychiatry* 70, 869-879.
- Yahata, N., Morimoto, J., Hashimoto, R., Lisi, G., Shibata, K., Kawakubo, Y., Kuwabara, H., Kuroda, M., Yamada, T., Megumi, F., Imamizu, H., Nanez, J.E., Sr., Takahashi, H., Okamoto, Y., Kasai, K., Kato, N., Sasaki, Y., Watanabe, T., Kawato, M., 2016. A small number of abnormal brain connections predicts adult autism spectrum disorder. *Nat Commun* 7, 11254.
- Yeo, B.T., Krienen, F.M., Sepulcre, J., Sabuncu, M.R., Lashkari, D., Hollinshead, M., Roffman, J.L., Smoller, J.W., Zollei, L., Polimeni, J.R., Fischl, B., Liu, H., Buckner, R.L., 2011. The organization of the human cerebral cortex estimated by intrinsic functional connectivity. *J Neurophysiol* 106, 1125-1165.

Reviewer #2 (Remarks to the Author):

I think this is a highly interesting article, which provides potentially important insights into ASD. However, there are various conceptual and technical issues that need to be addressed before it can be published.

1) There are very little details about the diffusion map embedding. My understanding is that the

approach has a couple of parameters. The authors should explain a little more about the approach and indicate clearly how the parameters were set.

2) Related to point (1), I wonder if the finding (in Figure 1) can be explained by simple global differences in connectivity between ASD and TD. From what I can tell, for ASD participants have lower (positive) gradient scores and higher (negative) gradient scores. Could this be explained by global shift (e.g., lower correlations) in ASD's 20K x 20K (thresholded) connectivity matrix? This kind of proportional thresholding (see van den Heuvel et al., NeuroImage, 2017) might lead to artifactual issues.

3) Similarly for the stepwise connectivity, was the connectivity thresholded? How was it thresholded? See previous point (2). Ditto for rich-club.

4) Given the existence various arbitrary parameters (e.g., threshold), the authors should provide evidence that different parameter settings do not affect the results of the paper.

5) Given the recent concerns about random field theory (Eklund et al., PNAS, 2016), surface FWER correction should use permutation tests.

6) Figure 3C is unclear. I don't quite understand the procedure. Please elaborate in the methods section. Step 1 presumably included A1, S1 and V1. Aren't S1/A1 and V1 at different locations in the figure? So there will be two dots in step 1?

7) In the computation of connectivity distance, how are negative correlations handled?

8) When performing permutation test for regression, the authors should control for site difference. In other words, there should only be permutation within sites, but not across sites.

9) As suggested by recent benchmarking (e.g., Ciric et al., NeuroImage, 2017), global signal regression (GSR) clearly outperforms CompCor in the removing motion-related artifacts. I suggest the authors utilize GSR instead of CompCor in the preprocessing.

Reviewer #3 (Remarks to the Author):

This report examines whether the flow of information from sensory systems to primarily the default system is atypical in ASD, and whether this represents an unitary brain mechanism for the disorder. The authors are applying concepts and tools that are newer to the field and so there probably needs to be a little more clarification on how these tools work and why they are doing what is being reported. It is true that some of this has been published elsewhere, but it was difficult to read without first reading several other papers. This aspect of the paper can be improved. At first blush the paper seems quite novel; however, I also wonder whether some of the results are just a reflection of things that have already been reported. Nonetheless, this was an interesting paper. More detailed comments are below.

First, based on everything we know about ASD. The various findings throughout the imaging literature, the variable genetics, behavior, etc, it's hard to square that what is being examined here would be the unitary brain mechanism for the disorder.

The entire paper is pitched as an issue in ASD as an issue with information flow from sensory areas to the higher order heteromodal areas. In this sense, the text is quite biased toward this idea without a closer look at the data. For example the authors note:

"The first principal gradient explained 24% of connectome variance in our dataset (with similar

variance explained in ASD and controls across gradients, $t=0.87$ $p>0.38$; Fig. S2), and showed a gradual transition from sensory networks towards the transmodal DMN (Fig. 1A)“ This is actually, not true. What the data is showing is that everything (i.e., all networks not just sensory) show a gradual transition to the DMN. This result, to me, doesn't actually fit the general theme the authors are trying to convey.

Along the same line, it is quite likely that this particular result in Figure 1 is simply secondary to the default network being negatively correlated with everything else. Thus, are the findings here related to the gradients actually telling us something different than what has already been shown in the literature?

With regard to the propagation of flow, it seems as if the result would be fundamentally different if you simply change the starting seed regions.

The entire rich club analysis was out of place and really was hard to link to the other data being presented. It's not clear what the motivation was for the analysis or why it would relate to the pieces being examined. There is also no attempts to relate these findings to what's already out there in the literature.

If you have two datasets why are you doing cross-validation for the predictions? Just generate the model on one set and test it on the other.

The discussion needs a lot of work. It's hard to follow, and a bit tangential to the actual findings in this report. There seems to be a disconnect of what is being shown here and the ASD literature that relates to it.

Regarding motion. I understand that the authors did extra work to handle motion related issues, but some of the methods shown here have been shown to be non-optimal for this purpose. Relying on mean frame displacement numbers doesn't remove the large spike in the data that are variable across these cohorts and are the major reason for the artifacts seen in them. In addition, regressing mean FD has also been shown to be non-optimal. It would be important I think to put in a few controls analyses here. For example comparing kids vs adults in the control should probably show a different pattern than the ASD comparison. If they don't then it could be a movement issue in the findings here. In addition, the authors can simply split the control population into high and low movers and look at the trends, do they resemble ASD comparisons?

RESPONSE TO REVIEWS (NCOMMS-18-10890)

We would like to thank the Editor for giving us the opportunity to resubmit a revised manuscript and we thank the Reviewers recognizing the novelty, quality, and potential impact of our work. We are grateful to the Reviewers for their constructive comments. We found the suggestions helpful for strengthening our conclusions, and we feel that the changes improved the quality of our revised manuscript. In particular, we have carried out all the additional control analyses (*i.e.*, correction for motion, global signal, overall connectivity shift; use of variable matrix thresholding, variable seed initialization for stepwise connectivity analysis, permutation-based cluster inference, and leave-one site out cross-validation) requested by Reviewer 2 and 3 and our main findings remained consistent. To address the comments of Reviewer 1 and those highlighted at the Editorial level, we now contextualize our findings in greater detail within the broader ASD literature, provide further details on the methodology, and address how our study provides a novel perspective on atypical functional network organization in autism. We have addressed all comments point-by-point and highlighted the corresponding changes in the main manuscript and supplementary materials in yellow.

REVIEWER #1

We thank Reviewer 1 for the evaluation of our work and the feedback.

1) Hong and colleagues report functional connectivity differences between young adults with ASD vs. age-matched controls. The authors downloaded preprocessed resting state fMRI data from the Autism Brain Imaging Data Exchange (ABIDE) initiative, which contains a heterogeneous accumulation of inconsistently acquired data from multiple institutions. The current report is based on a particular mode of resting state fMRI analysis, first described by Margulies et al. (2016), in which "diffusion embedding" is used to effect dimensionality reduction of covariance structures in manner similar to principal component analysis (PCA). The major claim here is that the magnitude of "gradient scores" (effectively, principal components) in the ASD group is systematically weaker than that obtained in controls. As illustrated in Figure 1, this lower magnitude appears as a modest reduction in functional connectivity strength. These results are interpreted as demonstrating "perturbed propagation of intrinsic flow from sensory towards transmodal areas" which finding is related to the "deficits in social cognition and low-level behavioral symptoms, but not problems in communication" that are said [inaccurately regarding impaired communication] to be characteristic of ASD.

To clarify, the latter association to behavioral deficits is based on our pattern learning analysis of the three ADOS symptom severity subscales (*i.e.*, social cognition, repetitive behaviors/interests, communication) provided with the ABIDE dataset. We have now provided clearer wording in the revised abstract P. 2.

"Supervised pattern learning demonstrated that hierarchical features predicted deficits in social cognition and low-level behavioral symptoms, but not communication-related symptoms."

And introduction P. 4

"Finally, supervised pattern learning successfully used hierarchy features to predict symptom severity in social cognition and repetitive behavioral symptom domains, as indexed by the ADOS, in individual patients."

2) The first major problem with this work is total failure of appropriate scholarship. The literature cited by the authors primarily concerns the behavioral characteristics of ASD.

Pointedly omitted is any mention of prior resting state fMRI studies comparing ASD to controls. A non-expert could easily be misled into assuming that this paper represents the first time that functional connectivity abnormalities have been associated with ASD. In fact, crossing "ASD" x "resting state fMRI" in PubMed currently returns 140 hits of which 7 are reviews. The history of resting state fMRI in ASD now becomes relevant. A substantial but massively contradictory literature on this topic had accumulated as of 2012 at which juncture it became clear that virtually all extant studies were corrupted by systematically greater head motion artifact in ASD (Deen and Pelphrey, 2012). Since then, most studies, including the present work, have taken care to control for this confounding factor.

However, a consistent pattern of functional connectivity changes associated with ASD has not emerged. Most recent studies have taken a data-driven approach in which classifiers of various design are used to identify discriminating functional connectivity features. Classifier accuracy ranges from 60% (Nielsen et al., 2013) or 67% (Abraham et al., 2017) to 91% (Chen et al., 2015), depending on the selectivity of the inclusion criteria. Clear focality is notably absent over-all. Thus, discriminating node pairs derived by unsupervised classification are widely distributed over the whole brain, e.g., (Heinsfeld et al., 2018; Plitt et al., 2015), possibility with an emphasis on more posterior regions, e.g., (Keown et al., 2013). Alternatively, multiple functional systems, e.g., "salience, default mode, frontotemporal, motor, and visual networks" are reported as affected (Uddin et al., 2013). Or, functional connectivity is said to be globally "idiosyncratic" (Nunes et al., 2018). Curiously, recent reports tend to emphasize over- as opposed to under-connectivity (Plitt et al., 2015; Yahata et al., 2016), i.e., precisely the inverse of what is here reported. This discrepancy is, of course, not discussed.

We thank the Reviewer for pointing out the need for a more thorough discussion of the previous resting-state fMRI literature, and for suggesting these important papers. In the revised discussion, we provide a more extensive contextualization of our findings with respect to previous resting-state work. In particular, we cover: *i*) the relative lack of convergence across previous studies, *ii*) the overall lack of focality in specific regions, *iii*) the interplay between over- and under-connectivity, and *iv*) effects of head motion in both introduction and discussion (*P. 3, 11-12*).

"Numerous rs-fMRI studies have focused on the mapping of connectional anomalies in ASD, but so far there is little agreement in the location of findings and nature of connectivity changes^{1,2,3,4,5}. These inconclusive patterns may be attributable to a number of factors, including differences across studies with respect to the specific autism populations studied (e.g., clinical subtype, age, symptom severity), the functional systems examined, the analytic approaches employed. The effects of MRI-related confounds such as head motion during image acquisition³ and inconsistent image processing procedures across the studies⁸ are also a potential sources of variation."

"The majority of the autism functional imaging literature has focused on the mapping of connectivity anomalies. Findings appear somewhat heterogeneous in location and even direction, with some studies reporting scattered connectivity increases in addition to prevailing and most frequently reported connectivity reductions^{1,2,3,4,5}. Inconsistent differences at the group level may also propagate down to the individual subject level, and classifiers designed to dissociate ASD from controls have provided rather variable accuracies, often below behavioral criteria^{5,9}."

Overall, we believe that the rather heterogeneous literature motivates studies assessing functional imbalances from a different perspective. In this revision, we now articulate why an analysis in a novel reference frame, based on two innovative connectomic approaches (SFC and gradient analysis), provides additional and important perspectives on ASD. Please see discussion *P. 11-12*.

"Collectively, these findings suggest that a broader ASD phenotype might arise from imbalances in functional architecture across individuals that are not purely anchored on spatial constraints but rather network-level features, a finding also supported by overall higher variability in autism functional network

organization than in controls¹⁰. Our proposed reference frame based on gradual hierarchy and stepwise connectivity analyses targeted such more overarching principles that could explain diverse behavioral phenotypes of ASD in a unified framework, as opposed to interrogating specific functional systems. By nature of our methods summarizing overall network anomalies in the compressed connectivity space, a part of the current findings may be seen as simply recapitulating what has been already found in previous studies (e.g., reduced connectivity between different transmodal nodes, such as the anterior and posterior DMN hubs^{11, 12} as well as in multiple sensory areas^{13, 14}). These findings, however, have so far remained somewhat fragmented and thus hard to integrate, while our approach could uncover an underlying backbone of all these seemingly heterogeneous anomalies and make more direct claims about organizational principles and their differences.”

3) The second major problem is the credulous adoption of a particular approach to functional connectivity analysis as described by Margulies et al., 2016. Allowing that that work makes a worthwhile point concerning the representation of function within the cerebral cortex of normal subjects, it does not follow that "principal gradient" decomposition is uniquely suited to discovering how ASD alters resting state functional connectivity. But that is the implication, since all other approaches are ignored. It should be noted that, as visualized on the brain surface, the first few "principal gradients" closely resemble the functional connectivity components previously obtained by others using related but more straightforward methods, e.g., (Lee et al., 2012; Power et al., 2011; Yeo et al., 2011). Thus, if, the present results have any relevance to ASD, similar findings should be obtained by more straightforward methods. Absent such a demonstration, the reported results simply reflect how ASD appears as viewed through a particular lens. Given the diffuse findings reported by others, including findings obtained using the same ABIDE dataset, it is predictable that "principal gradients" abnormalities would be found in ASD. But, it remains unclear that anything fundamental has been learned about ASD. In any case, the authors are obliged to reconcile their finding of weaker functional connectivity in ASD vis a vis much of the recent literature.

We thank the Reviewer for these considerations. We would first like to clarify that our work was targeting cortical macroscale organization not only via the gradient mapping techniques from work of Margulies and colleagues (2016), but also based on stepwise functional connectivity analyses popularized by Sepulcre and colleagues (2012). As such, an important strength of our paper is the use of these complementary approaches, developed from two different research groups, for the first time. Our synthesis thus brings together gradient mapping, a data-driven dimension reduction, with SFC, which uses *a-priori* selected seeds in primary sensory systems^{15, 16}. In *Figure 3*, we directly integrate findings from both analyses and we used features from both approaches in the prediction of ADOS symptoms in *Figure 4*. The revised introduction now further emphasizes complementarity of both techniques (*P. 4*):

“These techniques offer complementary ways to characterize hierarchical anomalies in ASD - gradient mapping provides an unsupervised dimension reduction technique that describes the underlying hierarchy, while SFC uses a-priori regions within sensory areas to characterize the specific transitions from peripheral to core regions.”

Notably, the combination of gradient and SFC techniques means that our approach is distinct from other data-driven clustering techniques that explicitly decompose the connectome into network communities/modules^{17, 18, 19}. In fact, both gradient and stepwise connectome analysis share the assumption that cortical areas can be organized along a ‘continuous’ axis of cortical organization, which is not the aim of clustering-approaches that discretize the connectome into separate communities. In this way, our combined analytic approach provides a unique insight into the possibility that macroscale hierarchy deficits are at the core of ASD. This is now outlined on *P. 3*:

“In contrast to clustering-based decompositions of the brain into discrete communities^{17, 18, 19}, the principal gradient describes a continuous transition from sensory to default mode networks (DMN) and corresponds to a spatial hierarchy established in earlier primate tract-tracing studies^{20, 21, 22}”

Our focus on a hierarchical frame of reference also provides a novel perspective on network perturbations in autism that extrapolates from specific regions and networks towards an approach that targets overall transitions between different functional systems. To illustrate this point, we have considered our findings in the context of low-level connectivity anomalies in ASD that were previously reported (with, however, somewhat inconsistent findings, see also the previous point of R1). We have now further discussed the original motivation of our approach (P. 11-12)

“Collectively, these findings suggest that a broader ASD phenotype might arise from imbalances in functional architecture across individuals that are not purely anchored on spatial constraints but rather network-level features, a finding also supported by overall higher variability in autism functional network organization than in controls¹⁰. Our proposed reference frame based on gradual hierarchy and stepwise connectivity analyses targeted such more overarching principles that could explain diverse behavioral phenotypes of ASD in a unified framework, as opposed to interrogating specific functional systems. By nature of our methods summarizing overall network anomalies in the compressed connectivity space, a part of the current findings may be seen as simply recapitulating what has been already found in previous studies (e.g., reduced connectivity between different transmodal nodes, such as the anterior and posterior DMN hubs^{11, 12} as well as in multiple sensory areas^{13, 14}). These findings, however, have so far remained somewhat fragmented and thus hard to integrate, while our approach could uncover an underlying backbone of all these seemingly heterogeneous anomalies and make more direct claims about organizational principles and their differences.”

REVIEWER #2

I think this is a highly interesting article, which provides potentially important insights into ASD. However, there are various conceptual and technical issues that need to be addressed before it can be published.

We thank Reviewer 2 for the positive evaluation of our work and their constructive comments.

1) There are very little details about the diffusion map embedding. My understanding is that the approach has a couple of parameters. The authors should explain a little more about the approach and indicate clearly how the parameters were set.

Many thanks for this suggestion. We have now added additional explanations and parameters to the methods (P. 14-15).

“We applied diffusion map embedding^{15, 23}, a nonlinear manifold learning approach, to identify principal gradient components explaining connectome variance in descending order (each of 1×20484). In brief, the algorithm estimates a low-dimensional embedding from a high-dimensional connectome matrix. In this space, cortical vertices that are strongly inter-connected by either many connections or few very strong connections are closer together, whereas vertices with only little or no inter-connectivity are farther apart. The name of this approach, which belongs to the family of graph Laplacians, derives from the equivalence of the Euclidean distance between points in the embedded space and the diffusion distance between probability distributions centered at those points. Compared other non-linear manifold learning techniques, the diffusion maps algorithm is relatively robust to noise and computationally inexpensive. Notably, the algorithm is controlled by a single parameter α , which controls the influence of density of sampling points on the manifold ($\alpha = 0$, maximal influence; $\alpha = 1$, no influence). In this study, we followed previous recommendation and set $\alpha=0.5$, a choice that retain the global relations between data points in the embedded space and that has been suggested to be relatively robust to noise in the connectivity matrix.”

2) Related to point (1), I wonder if the finding (in Figure 1) can be explained by simple global differences in connectivity between ASD and TD. From what I can tell, for ASD participants have lower (positive) gradient scores and higher (negative) gradient scores. Could this be

explained by global shift (e.g., lower correlations) in ASD's 20K x 20K (thresholded) connectivity matrix? This kind of proportional thresholding (see van den Heuvel et al., NeuroImage, 2017) might lead to artefactual issues.

As suggested, we have now added a control analysis that corrects for overall connectivity strength. Importantly, we still observe similar (even amplified) between-group differences in gradient and SFC, suggesting that network perturbations in ASD are not simply an epiphenomenon of a global connectivity shift. We have added these results to the revised manuscript (Fig. S10, P.6-7).

“Our results were also robust when controlling for a number of confounding features of rs-fMRI processing, including [...], whole-brain connectivity shift²⁴ (see Methods). [...] Results were also similar when controlling for average whole-brain connectivity strength (Fig. S10), suggesting no strong influence of overall connectivity shift²⁴. ”

“Findings were similar when [...] additionally controlling for average whole-brain connectivity strength (Fig. S10).”

The method for this control analysis is now outlined on P.17.

“b) Correction for overall connectivity strength. A recent study suggested a potential confounding effect of differences in overall connectivity strength on case-control network analyses²⁴. We thus repeated our main analyses after additionally correcting for average whole-brain connectivity. “

3) Similarly, for the stepwise connectivity, was the connectivity thresholded? How was it thresholded? See previous point (2). Ditto for rich-club.

To ensure method consistency, we employed the same thresholding as for the diffusion embedding analysis following the approach from Margulies et al¹⁵ for all other analyses (stepwise, rich-club, connectivity distance). Briefly, we z-transformed and thresholded the connectivity matrix, leaving only the top 10% of weighted connections per row. This is now clarified on P. 15.

“In our study, SFC was generated for each subject using the findwalks.m function from the Brain Connectivity Toolbox. We used the same thresholded matrix as input in the gradient mapping analysis.”

And P. 16.

“For computational efficiency, we subdivided the cortex into 1014 similarly-sized parcels²⁵, computed a weighted functional connectome (1014x1014), and applied the same thresholding as in the above analyses.”

“We binarized the above thresholded functional matrix, taking only positive connectivity and multiplied it with the resulting distance matrix. We finally calculated row-wise sums of this matrix and divided it by the overall node degree, which conceptually provides the average distance of significant functional connections for a given region.”

4) Given the existence various arbitrary parameters (e.g., threshold), the authors should provide evidence that different parameter settings do not affect the results of the paper.

The initial matrix threshold of 10% was identical to the one published in Margulies et al. 2016, which was based on Yeo et al. 2011. To verify robustness, we replicated the main findings (gradients, stepwise connectivity) also when using 5%, 15%, 20%, and 25% initial thresholding. The figure is now included as a new *Fig. S9*.

And the findings and methods provided on *P.6* (see also #2 to Reviewer 2).

“Our results were also robust when controlling for a number of confounding features of rs-fMRI processing, including connectivity matrix thresholding²⁶, [...] (see Methods). Specifically, altered gradient patterns in ASD remained virtually identical or were even amplified when [...] analyzing differently thresholded connectivity matrices (5-25%, Fig. S9).”

and *P. 17*.

“c) Matrix thresholding. To assess effects of matrix thresholding on our results, we repeated the gradient and SFC analyses at 5-25% thresholded matrices”

5) Given the recent concerns about random field theory (Eklund et al., PNAS, 2016), surface FWER correction should use permutation tests.

The revised manuscript now also presents surface-based gradient comparisons after permutation-based thresholding. These findings are now shown in the *Fig. S4*.

Supra-threshold permutation test based on cluster size (10000 iterations, cluster threshold=0.01)

And detailed on P.5.

“Similar findings, albeit with slightly lower significance levels, were obtained with non-parametric permutation tests²⁷ (Fig. S4; see Methods for details).”

And P 17.

“d) Non-parametric correction for multiple comparisons. We repeated surface-wide comparison of gradient scores using non-parametric [...].”

6) Figure 3C is unclear. I don't quite understand the procedure. Please elaborate in the methods section. Step 1 presumably included A1, S1, and V1. Aren't S1/A1 and V1 at different locations in the figure? So there will be two dots in step 1?

To clarify, Figure 3C represents each surface vertex as a point in the two-dimensional gradient space (spanned by Gradient 1 and 2) in controls (*left*) and autism (*right*). The coloring of the vertices is based on the cumulative step length (*i.e.*, the proportion of that vertex being activated across 200 steps), when simultaneously seeding from V1, S1, and A1. Superimposed arrow paths illustrate the evolution of the centroid of activated data points in 2D gradient space from the individual seeds, sampled at intervals of 20 steps. We hope that these clarifications help to better understand Figure 3C, and we have incorporated them in the revised figure legend, P. 7.

“C) Stepwise functional connectivity analysis (SFC), in gradient space. Points are colored with respect to cumulative steps when simultaneously seeding from V1, S1 and A1. Trajectories (sampled at every 20th step) illustrate the direct propagation from each of the primary sensory seeds to transmodal DMN in controls (left). ASD show an initially more rapid transition; however, trajectories deflect from a straight path and do not reach the DMN, even after 200 steps. Histogram on the right show point densities, weighted by the cumulative SFC. See also Videos S13-S15.”

7) In the computation of connectivity distance, how are negative correlations handled?

We apologize for missing an important detail in describing the computation of connectivity distance. As for the other approaches, connectivity distance was computed based on individual matrices thresholded at 10%. We then binarized individual matrices, taking only positive connectivity strength to compute degree centrality. To make a final connectivity distance map, this cortex-wide centrality measure was multiplied by a surface-based geodesic distance map. There was, thus, no negative connections affecting our analysis. We clarify this on P. 16.

“We binarized the above thresholded functional matrix, taking only positive connectivity and multiplied it with the resulting distance matrix. We finally calculated row-wise sums of this matrix and divided it by the overall node degree, which conceptually provides the average distance of significant functional connections for a given region.”

8) When performing permutation test for regression, the authors should control for site difference. In other words, there should only be permutation within sites, but not across sites.

The Reviewer is correct in assuming marked between-site differences in the current data (given that ABIDE represents a retrospective data-sharing initiatives with little *a-priori* homogenization). In the current work, we thus statistically corrected for age and site effects prior to permutations. As such, group comparison and prediction analyses were carried out based on age and site-corrected residual data. Please see P.17 for additional clarification.

“As in the main analysis, we statistically corrected for age and site prior to any permutation.”

9) As suggested by recent benchmarking (e.g., Ciric et al., NeuroImage, 2017), global signal regression (GSR) clearly outperforms CompCor in the removing motion-related artifacts. I suggest the authors utilize GSR instead of CompCor in the preprocessing.

Given controversy on GSR in the resting-state literature^{28, 29, 30, 31}, we did not perform GSR in our main analysis. Indeed, despite GSR showing desirable motion control, it has been suggested that it may alter the whole-brain correlation distribution and potentially induce between-network anti-correlations^{30, 32, 33}. To nevertheless address the Reviewer’s concern, we have now carried out separate gradient and SFC analyses after GSR. Findings remained stable after correcting for global mean signal. These findings are now presented on P. 6, 7 and in Fig S8.

“Specifically, altered gradient patterns in ASD remained virtually identical or were even amplified when using GSR-preprocessed data (Fig S8) [...]”

“Findings were similar when analyzing GSR-processed data (Fig S8) [...]”

Robustness of main findings with respect to global signal regression (GSR)

REVIEWER #3

This report examines whether the flow of information from sensory systems to primarily the default system is atypical in ASD, and whether this represents an unitary brain mechanism for the disorder. The authors are applying concepts and tools that are newer to the field and so there probably needs to be a little more clarification on how these tools work and why they are doing what is being reported.

It is true that some of this has been published elsewhere, but it was difficult to read without first reading several other papers. This aspect of the paper can be improved.

We thank the Reviewer for recognizing the novelty of the work. We have now added further clarifications to the gradient and SFC analyses (P. 14-15).

“We applied diffusion map embedding^{15, 23}, a nonlinear manifold learning approach, to identify principal gradient components explaining connectome variance in descending order (each of 1×20484). In brief, the algorithm estimates a low-dimensional embedding from a high-dimensional connectome matrix. In this space, cortical vertices that are strongly inter-connected by either many connections or few very strong connections are closer together, whereas vertices with only little or no inter-connectivity are farther apart. The name of this approach, which belongs to the family of graph Laplacians, derives from the equivalence of the Euclidean distance between points in the embedded space and the diffusion distance between probability distributions centered at those points. Compared to other non-linear manifold learning techniques, the diffusion maps algorithm is relatively robust to noise and computationally inexpensive. Notably, the algorithm is controlled by a single parameter α , which controls the influence of density of sampling points on the manifold ($\alpha = 0$, maximal influence; $\alpha = 1$, no influence). In this study, we followed

previous recommendation and set $\alpha=0.5$, a choice that retain the global relations between data points in the embedded space and that has been suggested to be relatively robust to noise in the connectivity matrix. “

And P.15.

“Stepwise functional connectivity (SFC) extends conventional seed-based connectivity analyses by detecting not only direct inter-regional but also indirect - yet meaningful - associations along successive connectivity steps. It counts the number of all paths that connect a given cortical region/vertex to the target region/vertex in a given length of connectivity distance. In other words, at a step-distance l , the degree of SFC at a target node i was computed by counting all possible paths between i and the seed that has an exact length of l . In our study, SFC was generated for each subject using the `findwalks.m` function from the Brain Connectivity Toolbox. We used the same thresholded matrix as input in the gradient mapping analysis. As the degree of SFC exponentially grows with increasing steps, we followed a previous approach³⁴ to standardize the value at each step by subtracting the whole brain mean of a SFC map and dividing by its standard deviation. Therefore, each SFC map represents a relative increase of connectivity degree across different step-distances. It has been shown that sensory-driven SFC maps in healthy individuals reach the transmodal DMN core after a certain number of steps, suggesting that the DMN may be at the top of a functional hierarchy¹⁵. For an integrative analysis, we put seed areas simultaneously in three representative primary sensory cortices, including visual (V1; MNI coordinate x, y, z : -14/10 [left/right], -78, 8), auditory (A1; -54/58, -14, 8) and somatosensory (S1; -42/38, -29, 65) areas. Notably to build a robust SFC degree map at the individual level, we first estimated a group-level template for the ASD and control groups. In the next step, we build individual SFC maps based on subject-specific connectomes but also guided by group-level SFC, by multiplying it with a subject SFC map at each step. This two-step group-guided approach was chosen to mitigate recursive SFC calculations that were observed at the individual level. In controls, SFC maps generally converged in the DMN after $l=200$ steps. In addition to visualizing SFC maps in ASD and controls, we estimated how many SFC steps were required to reach a certain level of hierarchy, and compared between groups using t -tests with multiple comparisons correction. This calculation was preceded by assigning functional communities¹⁵ to levels of hierarchy from a model initially proposed by Mesulam³⁵ (Fig. 2; level 1: sensory and somatomotor networks, level 2: dorsal attention, salience networks, level 3: fronto-parietal network, level 4: DMN). “

At first blush the paper seems quite novel; however, I also wonder whether some of the results are just a reflection of things that have already been reported. Nonetheless, this was an interesting paper.

We thank the Reviewer for recognizing the novelty of the work. In addition to providing further information on the individual methodological techniques (see previous comment), we now also provide a more extensive contextualization of our approach and findings in the broader resting-state literature in ASD in the introduction (P.3).

“Numerous *rs-fMRI* studies have focused on the mapping of connectional anomalies in ASD, but so far there is little agreement in the location of findings and nature of connectivity changes^{1, 2, 3, 4, 5}. These inconclusive patterns may be attributable to a number of factors, including differences across studies with respect to the specific autism populations studied (e.g., clinical subtype, age, symptom severity), the functional systems examined, the analytic approaches employed. The effects of MRI-related confounds such as head motion during image acquisition² and inconsistent image processing procedures across the studies⁸ are also a potential sources of variation.”

And discussion (P. 11-12).

“The majority of the autism functional imaging literature has focused on the mapping of connectivity anomalies. Findings appear somewhat heterogeneous in location and even direction, with some studies reporting scattered connectivity increases in addition to prevailing and most frequently reported connectivity reductions^{1, 2, 3, 4, 5}. Inconsistent differences at the group level may also propagate down to the individual subject level, and classifiers trying to dissociate ASD from controls have provided rather variable accuracies, often below behavioral criteria^{5, 9}. Collectively, these findings suggest that a broader ASD phenotype might arise from imbalances in functional architecture across individuals that are not purely anchored on spatial constraints but rather network-level features, a finding also supported

by overall higher variability in autism functional network organization than in controls¹⁰. Our proposed reference frame based on gradual hierarchy and stepwise connectivity analyses targeted such more overarching principles that could explain diverse behavioral phenotypes of ASD in a unified framework, as opposed to interrogating its specific functional systems. By nature of our methods summarizing overall network anomalies in the compressed connectivity space, a part of the current findings may be seen as simply recapitulating what has been already found in previous studies (e.g., reduced connectivity between different transmodal nodes, such as the anterior and posterior DMN hubs^{11, 12} as well as in multiple sensory areas^{13, 14}). These findings, however, have so far remained somewhat fragmented and thus hard to integrate, while our approach could uncover an underlying backbone of all these seemingly heterogeneous anomalies and make more direct claims about organizational principles and their differences.”

More detailed comments are below.

First, based on everything we know about ASD. The various findings throughout the imaging literature, the variable genetics, behavior, etc, it’s hard to square that what is being examined here would be the unitary brain mechanism for the disorder.

Thanks for this comment. We have now toned down the wording and replaced ‘unitary brain mechanism’ by ‘system-level imbalance’, also accounting for the possibility of other potential mechanisms discoverable in different disciplines. Please see the abstract P. 2.

“The current work examined whether these phenotypical patterns may relate to an overarching system-level imbalance – specifically a disruption in macroscale hierarchy affecting integration and segregation of unimodal and transmodal networks.”

And discussion, P.11.

“Collectively, our multi-method approach provides converging evidence for atypical connectome hierarchy as a system-level substrate of ASD, a finding that could be replicated across included datasets, parameter choices, and when controlling for various methodological confounds.”

The entire paper is pitched as an issue in ASD as an issue with information flow from sensory areas to higher order heteromodal areas. In this sense, the text is quite biased toward this idea without a closer look at the data. E.g. the authors note: “The first principal gradient explained 24% of connectome variance in our dataset (with similar variance explained in ASD and controls across gradients, $t=0.87$ $p>0.38$; Fig. S2), and showed a gradual transition from sensory networks towards the transmodal DMN (Fig. 1A)” This is actually, not true. What the data is showing is that everything (i.e., all networks not just sensory) show a gradual transition to the DMN. This result, to me, doesn’t actually fit the general theme the authors are trying to convey. We thank the Reviewer for this comment. We have now clarified the above statement on P. 4-5.

“The first principal gradient explained 24% of connectome variance in our dataset (with similar variance explained in ASD and controls across gradients, $t=0.87$ $p>0.38$; Fig. S2). It describes a gradual transition across the cortical surface, with sensory networks on the one end of the axis and the transmodal DMN on the other end (Fig. 1A and Fig S3), replicating recent findings in healthy adults¹⁵.”

We also appreciate the Reviewer’s point regarding interpretation of gradients, noting that others have interpreted them to reflect information flow. While this may be correct, the interpretation will require additional data (e.g., effective connectivity) to fully assess; as such, we have reviewed our manuscript to ensure that we did not employ that interpretation or speak beyond the data.

While the Reviewer makes the accurate observation that all networks show a gradual transition towards the DMN, there is still a specific distance of each of them from the DMN with respect to the principal gradient: sensory networks have lowest scores, followed by salience and dorsal attention, and finally frontoparietal, and the DMN. We have also added these specifications to the

revised results and added a new *Fig. S3*, which is also in line with the findings from Margulies et al. 2016 based on a separate cohort of healthy adults.

Along the same line, it is quite likely that this particular result in Figure 1 is simply secondary to the default network being negatively correlated with everything else. Thus, are the findings here related to the gradients actually telling us something different than what has already been shown in the literature?

We thank the Reviewer for this remark and have made additional clarifications to the discussion (P. 10), also contrasting gradient results to findings of network anti-correlations between the default mode network and other systems.

“We employed a connectome gradient mapping that places all cortical regions along a continuous axis of macroscale organization. This approach provides a complementary view point to conventional parcellation-based studies of macroscale brain organization and connectivity, and sidesteps the need to define discrete networks, such as those obtained through clustering techniques^{15,36}. In controls, these techniques revealed a continuous transition from low-level sensory systems on one end and transmodal regions in the DMN on the other, with intermediary networks in-between, as seen in prior studies in healthy adults²⁰. Notably, this gradient does not simply recapitulate network correlations/anticorrelations since the DMN is relatively close to some “task-positive” systems, such as the frontoparietal network, while it is further away from others, such as the sensory/motor cortex. Instead, connectome gradients offer a continuous topographic framework to analyze and visualize both the integration as well as the segregation of different cortical systems in terms of their functional connectivity.”

Please also note that we have carried out extensive control analyses, including additional correction for connectivity strength, GSR, and sub-splits based on motion, which all suggested consistency of the primary findings and thus dispel concerns that findings may be attributed to these confounds and low-level alterations.

In addition to its benefits for characterizing healthy brain organization, our findings provide novel and robust evidence in terms of gradient compression in ASD. It is noteworthy that the relatively lower location of the DMN in the principal gradient of ASD actually coincides mainly with decreased connectivity of DMN nodes to other rich-club nodes, rather than with increased connectivity to sensory networks, for example. As such, these findings offer a novel perspective on within- and between-network connectivity in ASD based on the gradient arrangement of the cortical mosaic.

With regard to the propagation of flow, it seems as if the result would be fundamentally different if you simply change the starting seed regions.

We would first like to clarify that our in approach, the seed coordinates were based on those from the SFC study by Sepulcre et al. 2012. To demonstrate robustness with respect to changes in coordinates, we varied the seed locations slightly (by selecting the parcel at the vicinity of the original seed) within V1, S1, and A1, and performed SFC analyses iteratively 10 times. SFC based

on these spatially jittered seeds resulted in consistent patterns *i.e.*, SFC propagation recapitulating the principal gradient in controls and showing perturbed propagation in ASD (*Fig. S11*)

We also initiated SFCs in centroids of three intermediary networks *i.e.* salience, fronto-parietal, and dorsal attention networks. Notably, these findings also showed delayed/perturbed propagation towards the DMN in ASD, suggesting a consistent imbalance in hierarchy even when initiating the SFC in intermediary networks (*Fig. S12*).

Stepwise connectivity profiles seeding at intermediary network areas

We have added these methods and results to the paper, *P. 7 and Fig. S11 and S12*.

“Furthermore, SFC patterns and group differences were consistent even when changing seed regions, away from the coordinates in primary sensory/somatosensory areas based on a previous study³⁷ to those in their vicinity, or even to hierarchically intermediary networks (Fig. S11, S12).”

The entire rich club analysis was out of place and really was hard to link to the other data being presented. It's not clear what the motivation was for the analysis or why it would relate to the pieces being examined. There are also no attempts to relate these findings to what's already out there in the literature.

In our revised manuscript, we have now provided a more compelling justification for the complementary perspective on atypical network hierarchy in ASD offered by the rich-club analysis. This analysis helps to conceptualize our findings also from a network topology perspective, thus linking our results to the graph theory literature. As suggested, we also discuss rich-club findings within the broader autism and network neuroscience literature now and also provide additional motivation for using this approach. Please see the revised introduction P. 4.

"We furthermore characterized gradient and SFC findings in ASD with graph theoretical metrics, specifically connectivity distance and rich-club organization^{38,39}, key topological features that offer a complementary viewpoint on cortical hierarchy i.e. the dissociation of a transmodal network core that aggregates most long-distance connections from peripheral networks with rather short-range connectivity. In this analysis, atypical gradients and stepwise connectivity in ASD related to a selective disruption in long-range connectivity, co-occurring with a deficit to fully activate the "rich-club"."

If you have two datasets why are you doing cross-validation for the predictions? Just generate the model on one set and test it on the other.

We thank the Reviewer for this suggestion. Firstly, please note that current work opted for a 5-fold cross-validation, which has been argued to result in more conservative estimates of generalization performance compared to commonly-used leave-one-out cross-validation or even within-sample correlations⁴⁰.

Secondly, while we agree with the Reviewer that one ultimate purpose of such tools would be cross-site prediction, it is important to emphasize that the ABIDE repositories were generated retrospectively, with little homogenization of the MRI data across sites (cohorts were scanned on both Siemens and Philipps 3T scanners using different sequences and scanning parameters). As we have shown previously at the level of structural MRI data⁷, this results in strong between-site differences in multiple MRI metrics. We believe that such cross-site heterogeneity unfortunately compromises the performance of statistical learners and may, in turn, necessitate statistical control for between-site effects (which can only be carried out adequately when training and test datasets include pooled data from different sites). To nevertheless address the Reviewer's concern, we have now evaluated performance of a leave-one-site-out prediction using the ABIDE-I dataset. Specifically, we trained using two of the three sites (*i.e.* NYU, PITT, USM) and used the remaining one for testing. While this classifier still performed significantly above chance levels, it achieved lower accuracy than when using 5-fold validation. These results are now shown on P.9.

"When predicting total using a leave-one-site-out strategy instead of 5-fold cross-validation, classifier performance had a lower accuracy but was still significantly above chance levels for total ADOS and social cognition subdomain scores (MAE=2.09/2.26, median r=0.29/0.30, permutation-test $p<0.003$). The prediction for communication and repeated behavior did not reach to the significant level (permutation-test $p>0.1$)."

We have further clarified our approach and outline the need for homogenized, prospective initiatives in which cross-site prediction may represent an important aim, please see P. 12.

"Given that these initiatives may also highlight variability in data quality across sites, they motivated us to perform an extensive case screening, several statistical corrections and validation experiments to confirm robustness of our findings. Specifically, we verified our results in variations of MRI quality and motion, given that such confounds have been suggested to also contribute to the marked heterogeneity across previous results in ASD². In contrast to homogenized multisite data, heterogeneous datasets such as ABIDE

may pose unique constraints on the accuracy of predictive routines that operate from one site to another. Indeed, a leave-one-site-out learner did not achieve as high accuracy as a 5-fold cross-validation learner that pooled data from sites. Notably, the employed 5-fold cross-validation nevertheless provides a more thorough control on classifier optimism compared to commonly applied within-sample regression or leave-one-out cross-validation.”

The discussion needs a lot of work. It’s hard to follow, and a bit tangential to the actual findings in this report. There seems to be a disconnect of what is being shown here and the ASD literature that relates to it.

As suggested, we now elaborate more on specific findings in the discussion and also embedded our results in the context of broader ASD literature. Please see *P. 11*.

“The majority of the autism functional imaging literature has focused on the mapping of connectivity anomalies. Findings appear somewhat heterogeneous in location and even direction, with some studies reporting scattered connectivity increases in addition to prevailing and most frequently reported connectivity reductions^{1, 2, 3, 4, 5}. Inconsistent differences at the group level may also propagate down to the individual subject level, and classifiers trying to dissociate ASD from controls have provided rather variable accuracies, often below behavioral criteria^{5, 9}. Collectively, these findings suggest that a broader ASD phenotype might arise from imbalances in functional architecture across individuals that are not purely anchored on spatial constraints but rather network-level features, a finding also supported by overall higher variability in autism functional network organization than in controls¹⁰. Our proposed reference frame based on gradual hierarchy and stepwise connectivity analyses targeted such more overarching principles that could explain diverse behavioral phenotypes of ASD in a unified framework, as opposed to interrogating its specific functional systems. By nature of our methods summarizing overall network anomalies in the compressed connectivity space, a part of the current findings may be seen as simply recapitulating what has been already found in previous studies (e.g., reduced connectivity between different transmodal nodes, such as the anterior and posterior DMN hubs^{11, 12} as well as in multiple sensory areas^{13, 14}). These findings, however, have so far remained somewhat fragmented and thus hard to integrate, while our approach could uncover an underlying backbone of all these seemingly heterogeneous anomalies and make more direct claims about organizational principles and their differences.”

Regarding motion. I understand that the authors did extra work to handle motion related issues, but some of the methods shown here have been shown to be non-optimal for this purpose. Relying on mean frame displacement numbers doesn’t remove the large spike in the data that are variable across these cohorts and are the major reason for the artifacts seen in them. In addition, regressing mean FD has also been shown to be non-optimal. It would be important I think to put in a few controls analyses here. For example, comparing kids vs adults in the control should probably show a different pattern than the ASD comparison. If they don’t then it could be a movement issue in the findings here. In addition, the authors can simply split the control population into high and low movers and look at the trends, do they resemble ASD comparisons?

To further dispel the potential effect of motion confounds on our results, we selected 50% of individuals with ASD with the lowest head motion as well as a motion-matched subgroup of 50% controls. Using this motion-minimal subset, we evaluated whether effect sizes of the contrast between ASD and controls are similar in the clusters of significant findings. We also performed an SFC analysis in this cohort, and assessed consistency of our main findings. The analyses confirmed a gradient compression in ASD and delayed/perturbed stepwise connectivity, despite the more extensive head motion screening.

Please see *P. 10*.

“Head motion is thought to be a major confound in fMRI studies, and to rule this out as an interpretation of our results, we conducted three additional analyses. Firstly, we repeated between-group comparisons

after including mean framewise displacement as an additional control covariate⁴¹. Secondly, we selected the 50% of individuals with ASD who had the lowest head motion and a motion-matched subgroup of 50% controls. Using this motion-minimized cohort, we carried out two analyses: 1) taking the gradient score from those original significant clusters and evaluating whether ASD still shows a meaningful group difference (i.e. effect size) compared to healthy controls, and 2) carrying out the SFC analysis and assessing if our original findings are replicated from this low-motion data. Finally, we correlated the individual z-scores of SFC anomalies with average framewise displacement, to assess whether degree of anomalies correlated with head motion estimates. None of these verification experiments indicated a noticeable impact of head motion on our findings. See **Fig. S16-18** for details.”

And the new *Figure S17*.

As suggested by the Reviewer, we also split our control population into the high and low movers (based on the 50% cutoff from the previous analysis) and compared gradient scores between cohort. Findings are shown below and in the new *Figure S18*; importantly, they do not resemble overall ASD vs. control differences, confirming a rather small impact of motion.

Finally, please note that we could also replicate our main findings after additionally correcting for global mean signal, an approach suggested to account for potential head motion confounds. Please see *Fig. S8 and P.6 and 7*.

REFERENCES FOR RESPONSE LETTER

1. Yahata N, *et al.* A small number of abnormal brain connections predicts adult autism spectrum disorder. *Nat Commun* **7**, 11254 (2016).
2. Deen B, Pelphrey K. Perspective: Brain scans need a rethink. *Nature* **491**, S20 (2012).
3. Keown Christopher L, Shih P, Nair A, Peterson N, Mulvey Mark E, Müller R-A. Local Functional Overconnectivity in Posterior Brain Regions Is Associated with Symptom Severity in Autism Spectrum Disorders. *Cell Reports* **5**, 567-572 (2013).
4. Uddin LQ. Idiosyncratic connectivity in autism: developmental and anatomical considerations. *Trends Neurosci* **38**, 261-263 (2015).
5. Heinsfeld AS, Franco AR, Craddock RC, Buchweitz A, Meneguzzi F. Identification of autism spectrum disorder using deep learning and the ABIDE dataset. *NeuroImage: Clinical* **17**, 16-23 (2018).
6. Lenroot R, Yeung PK. Heterogeneity within Autism Spectrum Disorders: What have We Learned from Neuroimaging Studies? *Frontiers in human neuroscience* **7**, (2013).
7. Hong S-J, Valk SL, Di Martino A, Milham MP, Bernhardt BC. Multidimensional Neuroanatomical Subtyping of Autism Spectrum Disorder. *Cerebral Cortex*, 1-11 (2017).
8. Abraham A, *et al.* Deriving reproducible biomarkers from multi-site resting-state data: An Autism-based example. *NeuroImage* **147**, 736-745 (2017).
9. Plitt M, Barnes KA, Martin A. Functional connectivity classification of autism identifies highly predictive brain features but falls short of biomarker standards. *Neuroimage Clin* **7**, 359-366 (2015).
10. Nunes AS, Peatfield N, Vakorin V, Doesburg SM. Idiosyncratic organization of cortical networks in autism spectrum disorder. *Neuroimage*, (2018).
11. Di Martino A, *et al.* The autism brain imaging data exchange: towards a large-scale evaluation of the intrinsic brain architecture in autism. *Mol Psychiatry* **19**, 659-667 (2014).
12. Joshi G, *et al.* Integration and Segregation of Default Mode Network Resting-State Functional Connectivity in Transition-Age Males with High-Functioning Autism Spectrum Disorder: A Proof-of-Concept Study. *Brain Connect* **7**, 558-573 (2017).
13. Villalobos ME, Mizuno A, Dahl BC, Kemmotsu N, Müller R-A. Reduced functional connectivity between V1 and inferior frontal cortex associated with visuomotor performance in autism. *NeuroImage* **25**, 916-925 (2005).
14. Khan S, *et al.* Somatosensory cortex functional connectivity abnormalities in autism show opposite trends, depending on direction and spatial scale. *Brain : a journal of neurology* **138**, 1394-1409 (2015).
15. Margulies DS, *et al.* Situating the default-mode network along a principal gradient of macroscale cortical organization. *Proceedings of the National Academy of Sciences of the United States of America* **113**, 12574-12579 (2016).
16. Sepulcre J, Sabuncu MR, Yeo TB, Liu H, Johnson KA. Stepwise connectivity of the modal cortex reveals the multimodal organization of the human brain. *The Journal of neuroscience : the official journal of the Society for Neuroscience* **32**, 10649-10661 (2012).
17. Lee MH, *et al.* Clustering of resting state networks. *PLoS one* **7**, e40370 (2012).
18. Power JD, *et al.* Functional network organization of the human brain. *Neuron* **72**, 665-678 (2011).
19. Yeo BT, *et al.* The organization of the human cerebral cortex estimated by intrinsic functional connectivity. *Journal of neurophysiology* **106**, 1125-1165 (2011).
20. Felleman DJ, Van Essen DC. Distributed hierarchical processing in the primate cerebral cortex. *Cereb Cortex* **1**, 1-47 (1991).
21. Hilgetag CC, O'Neill MA, Young MP. Hierarchical organization of macaque and cat cortical sensory systems explored with a novel network processor. *Philosophical transactions of the Royal Society of London Series B, Biological sciences* **355**, 71-89 (2000).
22. Markov NT, Ercsey-Ravasz M, Van Essen DC, Knoblauch K, Toroczkai Z, Kennedy H. Cortical high-density counterstream architectures. *Science* **342**, 1238406 (2013).
23. Coifman RR, *et al.* Geometric diffusions as a tool for harmonic analysis and structure definition of data: Diffusion maps. *Proceedings of the National Academy of Sciences of the United States of America* **102**, 7426-7431 (2005).
24. van den Heuvel MP, de Lange SC, Zalesky A, Seguin C, Yeo BTT, Schmidt R. Proportional thresholding in resting-state fMRI functional connectivity networks and consequences for patient-control connectome studies: Issues and recommendations. *NeuroImage* **152**, 437-449 (2017).
25. Cammoun L, *et al.* Mapping the human connectome at multiple scales with diffusion spectrum MRI. *Journal of neuroscience methods* **203**, 386-397 (2012).

26. van Wijk BC, Stam CJ, Daffertshofer A. Comparing brain networks of different size and connectivity density using graph theory. *PLoS one* **5**, e13701 (2010).
27. Nichols TE, Holmes AP. Nonparametric permutation tests for functional neuroimaging: a primer with examples. *Human brain mapping* **15**, 1-25 (2002).
28. Uddin LQ. Mixed Signals: On Separating Brain Signal from Noise. *Trends in cognitive sciences* **21**, 405-406 (2017).
29. Power JD, Laumann TO, Plitt M, Martin A, Petersen SE. On Global fMRI Signals and Simulations. *Trends in cognitive sciences* **21**, 911-913 (2017).
30. Murphy K, Fox MD. Towards a consensus regarding global signal regression for resting state functional connectivity MRI. *NeuroImage* **154**, 169-173 (2017).
31. Power JD, Plitt M, Laumann TO, Martin A. Sources and implications of whole-brain fMRI signals in humans. *NeuroImage*, (2016).
32. Saad ZS, *et al.* Trouble at Rest: How Correlation Patterns and Group Differences Become Distorted After Global Signal Regression. *Brain connectivity* **2**, 25-32 (2012).
33. S. AJ, Jason DT, Melissa L-L, Eun-Kee J, Krishnaji D, Deborah Y-T. Network anticorrelations, global regression, and phase-shifted soft tissue correction. *Human brain mapping* **32**, 919-934 (2011).
34. Carmona S, *et al.* Sensation-to-cognition cortical streams in attention-deficit/hyperactivity disorder. *Human brain mapping* **36**, 2544-2557 (2015).
35. Mesulam MM. From sensation to cognition. *Brain : a journal of neurology* **121 (Pt 6)**, 1013-1052 (1998).
36. Haak KV, Marquand AF, Beckmann CF. Connectopic mapping with resting-state fMRI. *NeuroImage* **170**, 83-94 (2018).
37. Sepulcre J, Liu H, Talukdar T, Martincorena I, Yeo BT, Buckner RL. The organization of local and distant functional connectivity in the human brain. *PLoS computational biology* **6**, e1000808 (2010).
38. van den Heuvel MP, Kahn RS, Goni J, Sporns O. High-cost, high-capacity backbone for global brain communication. *Proc Natl Acad Sci U S A* **109**, 11372-11377 (2012).
39. van den Heuvel MP, Sporns O. Rich-club organization of the human connectome. *The Journal of neuroscience : the official journal of the Society for Neuroscience* **31**, 15775-15786 (2011).
40. Arlot S, Celisse A. A survey of cross-validation procedures for model selection. *Statist Surv* **4**, 40-79 (2010).
41. Power JD, Barnes KA, Snyder AZ, Schlaggar BL, Petersen SE. Spurious but systematic correlations in functional connectivity MRI networks arise from subject motion. *NeuroImage* **59**, 2142-2154 (2012).

Reviewers' comments:

Reviewer #1 (Remarks to the Author):

The authors have addressed my principal concerns. Specifically, the scholarship now is adequate.

Reviewer #2 (Remarks to the Author):

The manuscript is largely improved. I only have one minor comment.

I would like to reframe reviewer 3's comment "Along the same line, it is quite likely that this particular result in Figure 1 is simply secondary to the default network being negatively correlated with everything else." as follows: is it possible that ASD participants have stronger FC between the default network and the remaining brain networks. This increased connectivity decreases the "distance" between the default network and other brain networks, the results of which are manifested in the SFC and gradient analysis.

To test this possibility, the authors can check if there is indeed stronger FC between the default network and other brain regions in ASD participants relative to TD participants. The author can also take the 20484 x 20484 FC matrix from TD participants and artificially increase connectivity between default network and other brain regions to see if the resulting matrix recapitulates the authors' results.

I should mention that even if the above hypothesis is true, I don't see it as necessarily detrimental to the paper, but instead it will strengthen the connection between the current study and previous case-control FC analyses.

Reviewer #3 (Remarks to the Author):

The authors go about responding to many of the concerns of the reviewers in the current version of the paper. I like the new approach and the goals here; however, there appears to be some disconnect with regard to some of the claims and responses. With that said others were thoroughly addressed. I'll run through a few here that not adequately addressed.

Two reviewers highlighted the lack of context and relationships of the current report to the existing literature on this front which is substantial. I was expecting a fairly major re-write and discussion regarding how all of this work related; yet, very little was actually added outside a half of a paragraph in the introduction, and the discussion is very superficial. Indeed there is an entire section in the paper on the rich club organization yet not one paper that has already done this in ASD is considered.

I had made an original comment regarding how everything (i.e., all networks not just sensory) show a gradual transition to the DMN and that the entire context of the paper doesn't make sense as currently written b/c it doesn't square with the data. The reviewer responds with "While the Reviewer makes the accurate observation that all networks show a gradual transition towards the DMN, there is still a specific distance of each of them from the DMN with respect to the principal gradient: sensory networks have lowest scores, followed by salience and dorsal attention, and finally frontoparietal, and the DMN." However, if you actually look at the figure they provide it still doesn't square. The salience network is at best no different than visual and, at worst, is more similar to sensory motor. In addition the fact that the frontoparietal network lands close to the DMN is likely simply because all of their regions are juxtaposed and share blurred signals.

I also pointed out why the findings here are likely simply the result of the default network being negatively correlated with everything else and that that dynamic in ASD is altered. Of course the later as been shown several times in the literature. There was no response or analysis conducted that would disprove this point. The only thing provided is a paragraph recapitulating the method and some discussion on denoising techniques.

I also had a comment regarding the result with regard to the propogation of information flow from sensory to DMN, which is the main thread of the paper, might change with different seed regions. In their supplementary analysis from what I can tell that seemed to be true. That the 'flow' goes from seed to DMN as expected. So, the main result here is secondary to where the seed started and changes the interpretation significantly.

I also commented on the potential for overfitting here and noted that because of the vast data available it would be wise to determine if the models generalize across datasets instead of simply cross validation. Indeed, the potential for generalization was one of the main arguments for the use of this dataset. The authors didn't do what I suggested. They did do a leave one out procedure which I did not suggest and I would not include. These models are unstable and unreliable. They did do a "cross-site" validation where the results were significantly weakened with some results now being non-significant. The authors make a valid point regarding cross site differences with regard to the modeling; however there are ways to handle this. Nonetheless, these results question the veracity of the findings in general.

I also pointed out the difficulty in following the discussion and that it needed to be re-written and focused. Very little was actually done, and thus my impression here remains the same.

[Redacted]

RESPONSE TO REVIEWS (NCOMMS-18-10890)

We would first like to thank the Editor for the constructive and balanced handling of our submission, and for giving us the opportunity to submit another revision. We thank all Reviewers for their comments and guidance. The revised manuscript now provides a more balanced account of the literature, and includes the two additional suggested analyses, which support our overall findings. Please find below a point-to-point answer with references to the revised passages in the main manuscript. All changes have been highlighted in yellow.

Reviewer #1 (Remarks to the Author):

The authors have addressed my principal concerns. Specifically, the scholarship now is adequate.

We thank Reviewer 1 for finding our revisions and the scholarship appropriate.

Reviewer #2 (Remarks to the Author):

The manuscript is largely improved. I only have one minor comment.

Many thanks for appreciating our revision and for the additional comment.

I would like to reframe reviewer 3's comment "Along the same line, it is quite likely that this particular result in Figure 1 is simply secondary to the default network being negatively correlated with everything else." as follows: is it possible that ASD participants have stronger FC between the default network and the remaining brain networks. This increased connectivity decreases the "distance" between the default network and other brain networks, the results of which are manifested in the SFC and gradient analysis.

To test this possibility, the authors can check if there is indeed stronger FC between the default network and other brain regions in ASD participants relative to TD participants. The author can also take the 20484 x 20484 FC matrix from TD participants and artificially increase connectivity between default network and other brain regions to see if the resulting matrix recapitulates the authors' results. I should mention that even if the above hypothesis is true, I don't see it as necessarily detrimental to the paper, but instead it will strengthen the connection between the current study and previous case-control FC analyses.

We thank the Reviewer for these suggestions. We first carried out the suggested simulations, by systematically increasing the connectivity strength between DMN and non-DMN areas from 0 to 0.3 in steps of 0.1, and each time re-generating the diffusion-embedding gradient map. As suggested, we simulated from the average connectivity matrix of healthy controls and qualitatively assessed its differences in gradient scores. As shown on the next page, increasing connectivity strength gradually rendered the whole-brain gradient pattern more contracted, showing a suppression at high order cortices and increase at lower level areas, the pattern that the Reviewer initially speculated.

Simulation analysis (increasing connectivity strength between DMN and non-DMN)

The Reviewers are thus correct that increased connectivity of the DMN to other networks may *theoretically* explain the observed pattern. To further evaluate this hypothesis with actual data, we carried out a post-hoc seed-based functional connectivity analysis from significant clusters of gradient differences between ASD and controls in the DMN (see the regions highlighted in *FIGURE 1C* of the manuscript). Notably, however, these findings do not support an increase in DMN connectivity to other networks, but rather a selective reduction in intrinsic functional connectivity within the DMN (see the **FIGURE** on the next page). Specifically, we observed reduced mPFC connectivity to medial prefrontal and parietal cortices, angular, and lateral temporal cortices as well as reduced PCC/PCU connectivity to medial parietal and anteromedial prefrontal cortices. These post-hoc results thus indicate that the gradient compression in ASD is not related to increased DMN functional connectivity to other networks but rather to selective disconnection within the DMN. Interestingly, predominant cortico-cortical underconnectivity was also observed when seeding from lower-level clusters showing gradient increases (OT/pMTG). We have added this figure and the corresponding description to the revised manuscript (**FIGURE S5** and *P.6*).

“Notably seed-based functional connectivity analysis centered on clusters of significant gradient alterations revealed a predominant pattern of connectivity reductions, not increases (Fig S5). Indeed, mPFC and PCU showed decreased connectivity predominantly to other DMN regions, while unimodal convergence regions such as OT and pMTG displayed reduced connectivity to primary sensory and somatomotor cortices.”

Reviewer #3 (Remarks to the Author):

1) Two reviewers highlighted the lack of context and relationships of the current report to the existing literature on this front which is substantial. I was expecting a fairly major rewrite and discussion regarding how all of this work related; yet, very little was actually added outside a half of a paragraph in the introduction, and the discussion is very superficial. Indeed, there is an entire section in the paper on the rich club organization yet not one paper that has already done this in ASD is considered.

As suggested per Reviewer consensus and at the Editorial level, we have reorganized and expanded our discussion to more broadly interpret our findings in the context of previous work. We also commented on our rich club findings, again in the context of previous literature (see **DISCUSSION** in the revised manuscript, P.10-12).

“The present results suggest a diminished segregation in ASD between sensory systems and unimodal convergence regions such as pMTG and OT^{1,2} on the one hand, and transmodal hubs such as mPFC and PCC/PCU on the other hand. In typically developing individuals, an important role of OT and pMTG in integration of feedforward and feedback streams has previously been recognized in the visual and auditory system, particularly for face¹ and language processing³. Consistent with the clinical impairment in these domains, ASD-related atypical activations of these regions have been highlighted in several task-based fMRI studies⁴. DMN core nodes, such as mPFC and PCC/PCU, are among the most consistently activated regions during self-referential and introspective cognition⁵ as well as other-oriented cognitive perspective taking operations including mentalizing⁶. Again, these processes are highly impaired in ASD⁷. Across these and other cortical nodes, a series of previous rs-fMRI studies have already reported a series of results indicative of functional connectivity alterations in ASD⁸. However, findings appear heterogeneous in location and direction. Indeed, earlier studies with moderate sample size predominantly focusing on regions in the DMN have largely reported underconnectivity in ASD relative to controls^{9,10}, while others reported some exceptions^{11,12}. These heterogeneous results can be in part attributed to diversity in study parameters, participant inclusion criteria, imaging acquisition, processing and data quality control^{13,14}. However, more recent studies focusing on macroscale networks in larger samples have reconciled seemingly discordant findings of disconnections by revealing the co-occurrence of ASD-related over-connectivity and under-connectivity (e.g.,¹⁵ and⁸ for reviews). This mosaic pattern appears to be specific to the functional circuit employed, with cortico-cortical networks largely described as hypo-connected in ASD in accordance with the current findings, while subcortico-cortical circuits appear hyperconnected, particularly between the thalamus and somatomotor cortex^{15,16,17,18,19}. In parallel, few recent studies supported the emerging concept that intrinsic functional connectivity networks in ASD may be idiosyncratically organized^{20,21,22,23}, where ASD expresses greater variability in functional network organization compared to controls, with stronger effects in default mode and somatomotor connectivity²¹. Collectively, these findings suggest that a broader ASD phenotype might arise from imbalances in functional architecture across individuals that are not purely anchored on spatial constraints but rather network-level features. Our proposed reference frame based on gradual connectome transitions and stepwise connectivity analyses targeted such overarching principles that could explain diverse behavioral phenotypes of ASD in a unified framework, as opposed to interrogating specific functional circuits. [...]”

“In addition to offering a novel perspective on connectional anomalies in autism, we incorporated a more conventional graph-theoretical rich club taxonomy as well as spatial geodesic distance into our analytical stream. Using this approach, we found that rich club nodes showed the strongest reduction in gradient scores and observed more marked reductions in areas with long-range connectivity compared to those with short-range functional connectivity profiles. Notably, our study derived a rich club backbone from both controls and ASD to stratify differences in gradient and SFC analyses, while other previous studies on rich club architecture first identified this subnetwork in ASD and controls separately, then compared their overall connectivity and spatial configuration between the groups^{24,25}. As such a direct link of findings in ours and previous work may not be straightforward. Yet, our findings are consistent with other reports of decreased network centrality, an alternative measure of node influence, across multiple cortical regions in individuals with ASD^{26,27}. Overall our results are in parallel to those previous studies that ASD is characterized by spatial and topology-level reorganization of this core subnetwork. Thus, beyond recapitulating previous and seemingly fragmented findings including reduced cortico-cortical functional connectivity between transmodal and sensorimotor networks^{28,29,30,31}, our approach provides more direct measures of atypical connectome cortical hierarchy as a system-level substrate of ASD. The availability of such objective markers will allow for further examinations across disorders to identify specific and transdiagnostic atypicalities, an effort that is becoming closer to reach with the sharing of transdiagnostic samples, such as the healthy brain network³².”

2) I had made an original comment regarding how everything (i.e., all networks not just sensory) show a gradual transition to the DMN and that the entire context of the paper doesn't make sense as currently written b/c it doesn't square with the data. The reviewer responds with

“While the Reviewer makes the accurate observation that all networks show a gradual transition towards the DMN, there is still a specific distance of each of them from the DMN with respect to the principal gradient: sensory networks have lowest scores, followed by salience and dorsal attention, and finally frontoparietal, and the DMN.” However, if you actually look at the figure they provide it still doesn’t square. The salience network is at best no different than visual and, at worst, is more similar to sensory motor. In addition the fact that the frontoparietal network lands close to the DMN is likely simply because all of their regions are juxtaposed and share blurred signals.

[Redacted]

We thank Reviewers for encouraging us to provide additional clarification regarding the overall approach. First, as also suggested by Reviewer 1, across multiple sections of the revised manuscript we have clarified the differentiation of our gradient mapping (based on the approach of Margulies et al., 2016) from the boundary detection framework championed by the Washington University Group^{33,34}. See **INTRODUCTION** at *P.3*:

“In contrast to clustering-based decompositions of the brain into discrete communities^{35,36} or recently developed connectivity boundary mapping techniques^{33,34,37}, cortex-wide gradient mapping techniques describe a continuous coordinate system”

And **DISCUSSION** at *P.11*.

“Notably this approach provides a complementary viewpoint to conventional parcellation-based studies of macroscale brain organization and connectivity, and sidesteps the need to define discrete networks, such as those obtained through clustering techniques^{35, 36} or through recently developed boundary mapping techniques that delineate rapid transitions between connectivity patterns of different cortical points^{33, 34, 37}. Although our method also leverages information of connectional affinity between areas similar to previous parcellation studies, the gradient mapping further projects these into a non-linear diffusion space and identifies principal components that describe main spatial axes in connectivity variations at the cortex-wide level. Therefore, the resultant gradient scores do not simply recapitulate network correlations/anticorrelations, as seen in their patterns of the DMN network that is relatively close to some “task-positive” systems (e.g., the frontoparietal network), while it is further away from others, such as the sensory/motor cortex.”

As also suggested by Reviewer 2, we have shown ASD and controls participants separately in **FIGURE S3** and rephrased the respective passage on P. 5.

“[...] showed a gradual axis of connectivity variations that placed low-level sensory systems on the one end and the transmodal DMN on the other end, with intermediary networks in-between, replicating recent data in healthy adults²⁰.”

3) I also pointed out why the findings here are likely simply the result of the default network being negatively correlated with everything else and that that dynamic in ASD is altered. Of course the later as been shown several times in the literature. There was no response or analysis conducted that would disprove this point. The only thing provided is a paragraph recapitulating the method and some discussion on denoising techniques.

[Redacted]

We have now generated functional connectivity profiles from DMN clusters in ASD and controls and demonstrated that DMN clusters showed significantly decreased functional connectivity to other DMN regions but not to every other network. These findings are shown in the **FIGURE**

above [See Response to Reviewer 2]. We also like to thank the Reviewer 2 for suggesting an addition control simulation, which we have now performed and which indicated that the findings do not relate to connectivity increases between the DMN and other networks but rather to connectivity reductions within the DMN itself. For details, please see the response to Reviewer 2.

4) I also had a comment regarding the result with regard to the propagation of information flow from sensory to DMN, which is the main thread of the paper, might change with different seed regions. In their supplementary analysis from what I can tell that seemed to be true. That the ‘flow’ goes from seed to DMN as expected. So, the main result here is secondary to where the seed started and changes the interpretation significantly

[Redacted]

Reviewer 2 is correct that SFC across these variable seed locations are largely converging, which supports our conclusions. We, nevertheless became aware of the need to reword the passages in **METHODS** and **RESULTS** (P. 18 and P.7 of the revised manuscript) to improve clarity.

Methods:

“Main SFC analyses were based on previously published coordinates³⁸. To dispel an influence of the exact seed location, we systematically varied seed coordinates. To first assess effects of slight variations in the initial seed coordinates, we identified immediate neighboring parcels of the V1, S1 and A1 areas (within a radius of 3mm; black dots in Fig. S12) and randomly selected three seeds from those neighbors, each from either V1 or S1 or A1, and conducted a SFC analysis for reproducibility. We repeated this procedure 10 times. To then evaluate SFC initialized from non-DMN networks, we also placed seeds in frontoparietal, salience, and dorsal attention networks (Fig. S13).”

Results:

“Furthermore, SFC patterns and group differences were consistent even when changing the initial seed regions to coordinates to those in the vicinity of the sensory seeds published in earlier work³⁸; notably, findings were even similar when seeding from intermediary, non-DMN, networks (Fig. S12, S13).”

As suggested by Reviewers 3 and 1, we have specifically deleted instances of ‘flow’ and ‘propagation’ when talking about the step wise connectivity patterns.

As suggested by Reviewer 2, we also increased the size of the bar charts in **FIGURE 2**.

5) I also commented on the potential for overfitting here and noted that because of the vast

data available it would be wise to determine if the models generalize across datasets instead of simply cross validation. Indeed, the potential for generalization was one of the main arguments for the use of this dataset. The authors didn't do what I suggested. They did do a leave one out procedure which I did not suggest and I would not include. These models are unstable and unreliable. They did do a "cross-site" validation where the results were significantly weakened with some results now being non-significant. The authors make a valid point regarding cross site differences with regard to the modeling; however, there are ways to handle this. Nonetheless, these results question the veracity of the findings in general.

[Redacted]

We now further clarified the approach on P. 9 of the revised manuscript. The approach is as Reviewer 2 understood it: we performed a *i)* 5-fold cross validation and *ii)* a leave-one-site-out cross-validation. We did not employ any leave-one-(subject)-out cross-validation analyses in the manuscript. In regard to the weakening of cross-site validation results, we would like to point to earlier studies showing that generally larger samples are needed to reach significant and generalizable accuracy³⁹. Together with source heterogeneity across different sites, sample size may have affected the reduced significance in the cross-site prediction accuracy.

"Using 5-fold cross-validations (where the classifier is repeatedly trained on 4 folds of the data and tested on the 5th fold), we found gradient and SFC features to significantly predict total ADOS scores [...]. Consistent with prior findings suggesting that cross-validated accuracy is affected by sample size³⁹, classifier accuracy decreased but was still above chance for total ADOS and social cognition subdomain scores when predicting total ADOS using a leave-one-site-out strategy instead of 5-fold cross-validation [...]"

I also pointed out the difficulty in following the discussion and that it needed to be re-written and focused. Very little was actually done, and thus my impression here remains the same.

[Redacted]

Our discussion now includes a more detailed account of previous functional connectivity findings in autism and a contextualization of our findings within this literature. See P. 10-12.

On P. 12, we furthermore revised several sections for clarity, and expanded on how our results may help contribute to our growing understanding of ASD pathophysiology.

"At the microcircuit level, atypical hierarchical organization may be related to anomalies in local signaling, specifically perturbations in the balance of excitation and inhibition (E/I). In Shank3 mice, an ASD model with established E/I imbalance related to mutations in synaptic scaffolding, recent work has shown reduced prefrontal functional connectivity to other higher order regions that was predictive of intellectual disability and socio-communicative impairments⁴⁰. Similar effects were demonstrated a different model related to mutations in the cell adhesion molecule CNTNAP2, where reductions in long-range rsfMRI connectivity particularly affected

heteromodal fronto-posterior components of the mouse DMN, an effect that was associated with reduced social investigation⁴¹. Some other molecular studies have also highlighted genes with effects on GABA/Glutamate pathways in ASD⁴² and associated to the broader behavioral phenotype that includes both sensory anomalies, as well as atypical social interactions^{43,44}. These studies collectively reinforce the concept that such imbalances may serve as a fundamental pathophysiological mechanism of ASD⁴⁵.”

REFERENCES FOR RESPONSE LETTER

1. Hocking J, Price CJ. The role of the posterior superior temporal sulcus in audiovisual processing. *Cereb Cortex* **18**, 2439-2449 (2008).
2. Kanwisher N, McDermott J, Chun MM. The fusiform face area: a module in human extrastriate cortex specialized for face perception. *J Neurosci* **17**, 4302-4311 (1997).
3. Wilson SM, Bautista A, McCarron A. Convergence of spoken and written language processing in the superior temporal sulcus. *Neuroimage* **171**, 62-74 (2018).
4. Hubbard AL, McNealy K, Zeeland AASV, Callan DE, Bookheimer SY, Dapretto M. Altered integration of speech and gesture in children with autism spectrum disorders. *Brain Behav* **2**, 606-619 (2012).
5. Buckner RL, Andrews-Hanna JR, Schacter DL. The brain's default network: anatomy, function, and relevance to disease. *Ann N Y Acad Sci* **1124**, 1-38 (2008).
6. Mitchell JP, Banaji MR, Macrae CN. The link between social cognition and self-referential thought in the medial prefrontal cortex. *Journal of cognitive neuroscience* **17**, 1306-1315 (2005).
7. Di Martino A, Ross K, Uddin LQ, Sklar AB, Castellanos FX, Milham MP. Functional Brain Correlates of Social and Nonsocial Processes in Autism Spectrum Disorders: An Activation Likelihood Estimation Meta-Analysis. *Biol Psychiat* **65**, 63-74 (2009).
8. Picci G, Gotts SJ, Scherf KS. A theoretical rut: revisiting and critically evaluating the generalized under/over-connectivity hypothesis of autism. *Dev Sci* **19**, 524-549 (2016).
9. Just MA, Cherkassky VL, Keller TA, Minshew NJ. Cortical activation and synchronization during sentence comprehension in high-functioning autism: evidence of underconnectivity. *Brain* **127**, 1811-1821 (2004).
10. Kennedy DP, Courchesne E. The intrinsic functional organization of the brain is altered in autism. *Neuroimage* **39**, 1877-1885 (2008).
11. Lynch CJ, Uddin LQ, Supekar K, Khouzam A, Phillips J, Menon V. Default mode network in childhood autism: posteromedial cortex heterogeneity and relationship with social deficits. *Biol Psychiatry* **74**, 212-219 (2013).
12. Supekar K, *et al.* Brain Hyperconnectivity in Children with Autism and its Links to Social Deficits. *Cell Reports* **5**, 738-747 (2013).
13. Nair A, Keown CL, Datko M, Shih P, Keehn B, Muller RA. Impact of methodological variables on functional connectivity findings in autism spectrum disorders. *Hum Brain Mapp* **35**, 4035-4048 (2014).
14. Müller R-A, Shih P, Keehn B, Deyoe JR, Leyden KM, Shukla DK. Underconnected, but How? A Survey of Functional Connectivity MRI Studies in Autism Spectrum Disorders. *Cerebral Cortex* **21**, 2233-2243 (2011).
15. Di Martino A, *et al.* Unraveling the miswired connectome: a developmental perspective. *Neuron* **83**, 1335-1353 (2014).

16. Cerliani L, Mennes M, Thomas RM, Di Martino A, Thioux M, Keyzers C. Increased Functional Connectivity Between Subcortical and Cortical Resting-State Networks in Autism Spectrum Disorder. *JAMA psychiatry* **72**, 767-777 (2015).
17. Chen JA, Penagarikano O, Belgard TG, Swarup V, Geschwind DH. The emerging picture of autism spectrum disorder: genetics and pathology. *Annu Rev Pathol* **10**, 111-144 (2015).
18. Tomasi D, Volkow ND. Reduced Local and Increased Long-Range Functional Connectivity of the Thalamus in Autism Spectrum Disorder. *Cereb Cortex*, (2017).
19. Woodward ND, Giraldo-Chica M, Rogers B, Cascio CJ. Thalamocortical Dysconnectivity in Autism Spectrum Disorder: An Analysis of the Autism Brain Imaging Data Exchange. *Biological Psychiatry: Cognitive Neuroscience and Neuroimaging* **2**, 76-84 (2017).
20. Hahamy A, Behrmann M, Malach R. The idiosyncratic brain: distortion of spontaneous connectivity patterns in autism spectrum disorder. *Nat Neurosci* **18**, 302-309 (2015).
21. Nunes AS, Peatfield N, Vakorin V, Doesburg SM. Idiosyncratic organization of cortical networks in autism spectrum disorder. *Neuroimage*, (2018).
22. Dickie EW, *et al.* Personalized Intrinsic Network Topography Mapping and Functional Connectivity Deficits in Autism Spectrum Disorder. *Biol Psychiatry* **84**, 278-286 (2018).
23. Keehn B, Muller RA, Townsend J. Atypical attentional networks and the emergence of autism. *Neurosci Biobehav Rev* **37**, 164-183 (2013).
24. Ray S, *et al.* Structural and functional connectivity of the human brain in autism spectrum disorders and attention-deficit/hyperactivity disorder: A rich club-organization study. *Hum Brain Mapp* **35**, 6032-6048 (2014).
25. Keown CL, Datko MC, Chen CP, Maximo JO, Jahedi A, Müller R-A. Network organization is globally atypical in autism: A graph theory study of intrinsic functional connectivity. *Biological psychiatry Cognitive neuroscience and neuroimaging* **2**, 66-75 (2017).
26. Lee Y, Park BY, James O, Kim SG, Park H. Autism Spectrum Disorder Related Functional Connectivity Changes in the Language Network in Children, Adolescents and Adults. *Front Hum Neurosci* **11**, (2017).
27. Di Martino A, *et al.* Shared and distinct intrinsic functional network centrality in autism and attention-deficit/hyperactivity disorder. *Biol Psychiatry* **74**, 623-632 (2013).
28. Di Martino A, *et al.* The autism brain imaging data exchange: towards a large-scale evaluation of the intrinsic brain architecture in autism. *Mol Psychiatry* **19**, 659-667 (2014).
29. Joshi G, *et al.* Integration and Segregation of Default Mode Network Resting-State Functional Connectivity in Transition-Age Males with High-Functioning Autism Spectrum Disorder: A Proof-of-Concept Study. *Brain Connect* **7**, 558-573 (2017).
30. Villalobos ME, Mizuno A, Dahl BC, Kemmotsu N, Müller R-A. Reduced functional connectivity between V1 and inferior frontal cortex associated with visuomotor performance in autism. *Neuroimage* **25**, 916-925 (2005).
31. Khan S, *et al.* Somatosensory cortex functional connectivity abnormalities in autism show opposite trends, depending on direction and spatial scale. *Brain* **138**, 1394-1409 (2015).
32. Alexander LM, *et al.* An open resource for transdiagnostic research in pediatric mental health and learning disorders. *Sci Data* **4**, 170181 (2017).

33. Cohen AL, *et al.* Defining functional areas in individual human brains using resting functional connectivity MRI. *NeuroImage* **41**, 45-57 (2008).
34. Gordon EM, Laumann TO, Adeyemo B, Huckins JF, Kelley WM, Petersen SE. Generation and Evaluation of a Cortical Area Parcellation from Resting-State Correlations. *Cereb Cortex* **26**, 288-303 (2016).
35. Power JD, *et al.* Functional network organization of the human brain. *Neuron* **72**, 665-678 (2011).
36. Yeo BT, *et al.* The organization of the human cerebral cortex estimated by intrinsic functional connectivity. *J Neurophysiol* **106**, 1125-1165 (2011).
37. Wig GS, *et al.* Parcellating an Individual Subject's Cortical and Subcortical Brain Structures Using Snowball Sampling of Resting-State Correlations. *Cerebral Cortex (New York, NY)* **24**, 2036-2054 (2014).
38. Sepulcre J, Sabuncu MR, Yeo TB, Liu H, Johnson KA. Stepwise connectivity of the modal cortex reveals the multimodal organization of the human brain. *J Neurosci* **32**, 10649-10661 (2012).
39. Moradi E, Khundrakpam B, Lewis JD, Evans AC, Tohka J. Predicting symptom severity in autism spectrum disorder based on cortical thickness measures in agglomerative data. *Neuroimage* **144**, 128-141 (2017).
40. Pagani M, *et al.* Deletion of autism risk gene Shank3 disrupts prefrontal connectivity. *bioRxiv*, (2018).
41. Liska A, *et al.* Homozygous Loss of Autism-Risk Gene CNTNAP2 Results in Reduced Local and Long-Range Prefrontal Functional Connectivity. *Cereb Cortex* **28**, 1141-1153 (2018).
42. Mariani J, *et al.* FOXP1-Dependent Dysregulation of GABA/Glutamate Neuron Differentiation in Autism Spectrum Disorders. *Cell* **162**, 375-390 (2015).
43. Bourgeron T. From the genetic architecture to synaptic plasticity in autism spectrum disorder. *Nat Rev Neurosci* **16**, 551-563 (2015).
44. Guilmatre A, Huguet G, Delorme R, Bourgeron T. The emerging role of SHANK genes in neuropsychiatric disorders. *Dev Neurobiol* **74**, 113-122 (2014).
45. Robertson CE, Baron-Cohen S. Sensory perception in autism. *Nat Rev Neurosci* **18**, 671-684 (2017).

REVIEWERS' COMMENTS:

Reviewer #1 (Remarks to the Author):

1. The scholarship of this work has been very substantially improved since the original submission. The added paragraphs detail the hugely contradictory nature of the current resting state fMRI literature on ASD. The added literature review has value.
2. Much of that literature is marked by poor technique and inadequate removal of artifact. This work is technically very good.
3. However, I remain skeptical regarding the magnitude of the reported ASD vs. control effects. Fig. 1A suggests that diffusion embedding gradients are weaker in ASD. But Figs. 1B and S3 raise doubts. I would suggest that moving Fig. S3 to main text would provide a more balanced view of the data.

Reviewer #2 (Remarks to the Author):

I am R2 in the current revision. The authors have fully addressed my concerns, including my comments about whether the gradient differences are simply a result of increased DMN's connectivity to other networks.

RESPONSE TO REVIEWS (NCOMMS-18-10890B)

We would first like to thank the Editor for the constructive and balanced handling of our submission, and for conditionally accepting our paper. We thank all Reviewers for their comments and guidance, and for being satisfied with our previous revisions.

REVIEWER 1

1. The scholarship of this work has been very substantially improved since the original submission. The added paragraphs detail the hugely contradictory nature of the current resting state fMRI literature on ASD. The added literature review has value.

We thank the Reviewer for appreciating the literature review of our paper.

2. Much of that literature is marked by poor technique and inadequate removal of artifact. This work is technically very good.

We thank the Reviewer for appreciating our methodology.

3. However, I remain skeptical regarding the magnitude of the reported ASD vs. control effects. Fig. 1A suggests that diffusion embedding gradients are weaker in ASD. But Figs. 1B and S3 raise doubts. I would suggest that moving Fig. S3 to main text would provide a more balanced view of the data.

We thank the Reviewer for this suggestion. We incorporated a slightly enhanced version of Figure S3 into Figure 1.

REVIEWER 2

1. I am R2 in the current revision. The authors have fully addressed my concerns, including my comments about whether the gradient differences are simply a result of increased DMN's connectivity to other networks.

We thank the Reviewer for the helpful and constructive comments.